# 1  How long do satellites need to overlap? Evaluation of climate data
# 2  stability from overlapping satellite records

Elizabeth C. Weatherhead[1], Jerald Harder[2], Eduardo A. Araujo-Pradere[3],
Greg Bodeker[4], Jason M. English[5], Lawrence E. Flynn[6], Stacey M. Frith[7], Jeffrey K Lazo[8], Peter
Pilewskie[2], Mark Weber[9], Thomas N. Woods[2]
[1] University of Colorado, Boulder, Colorado, USA
[2] Laboratory for Atmosphere and Space Physics, University of Colorado, Boulder, Colorado, USA
[3] School of Science, Miami Dade College, Miami, Florida, USA
[4] Bodeker Scientific, Alexandra, New Zealand
[5] Cooperative Institute for Research in Environmental Sciences, University of Colorado, Boulder, Colorado, USA /
NOAA Earth System Research Laboratory, Global Systems Division, 325 Broadway, Boulder, CO 80305
[6] NOAA, NESDIS, College Park, Maryland, USA
[7] Science Systems and Applications, Inc., Lanham, MD USA
[8] National Center for Atmospheric Research, Boulder, Colorado, USA
[9] University of Bremen FB1, Bremen, Germany
*Correspondence to:* Elizabeth Weatherhead (Betsy.Weatherhead@Colorado.edu)
**Abstract.** Sensors on satellites provide unprecedented understanding of the Earth's climate system by measuring
incoming solar radiation, as well as both passive and active observations of the entire Earth with outstanding spatial
and temporal coverage. A common challenge with satellite observations is to quantify their ability to provide well-
calibrated, long-term, stable records of the parameters they measure. Ground-based intercomparisons offer some
insight, while reference observations and internal calibrations give further assistance for understanding long-term
stability. A valuable tool for evaluating and developing long-term records from satellites is the examination of data
from overlapping satellite missions. This paper addresses how the length of overlap affects the ability to identify an
offset or a drift in the overlap of data between two sensors. Ozone and temperature datasets are used as examples
showing that overlap data can differ by latitude and can change over time. New results are presented for the general
case of sensor overlap by using SORCE SIM and SOLSTICE solar irradiance data as an example. To achieve a 1%
uncertainty in estimating the offset for these two instruments' measurement of the Mg II core (280 nm) requires
approximately 5 months of overlap. For relative drift to be identified within 0.1% per year uncertainty (0.00008
watts $m^{-2}$ $nm^{-1}$ $yr^{-1}$, the overlap for these two satellites would need to be 2.5 years. Additional overlap of satellite
measurements is needed if, as is the case for solar monitoring, unexpected jumps occur adding uncertainty to both
offsets and drifts; the additional length of time needed to account for a single jump in the overlap data may be as
large as 50% of the original overlap period in order to achieve the same desired confidence in the stability of the
merged dataset. Results presented here are directly applicable to satellite Earth observations. Approaches for Earth
observations offer additional challenges due to the complexity of the observations but Earth observations may also
benefit from ancillary observations taken from ground-based and in situ sources. Difficult choices need to be made
when monitoring approaches are considered; we outline some attempts at optimizing networks based on economic
principles. The careful evaluation of monitoring overlap is important to the appropriate application of observational
resources and to the usefulness of current and future observations.

*Keywords:* Satellite overlap, satellite monitoring, instrument intercomparison, instrument stability, climate records,
trend detection, homogenization of datasets, solar spectral irradiance, ozone.

**1. Introduction**
Stable, long-term time series of environmental data are critical to the ongoing investigation and understanding of the
environment. One of the fundamental requirements for construction of long-duration climate records is the ability to
analytically assess the characteristics of time series of different sensors so they can be combined into a single
reliable record. This need is particularly valuable for satellite-borne sensors that are susceptible to a wide variety of
sensitivity degradation mechanisms influenced by the space environment as well as by spacecraft, instrument, and
operational considerations. Many of these influences can contribute in unexpected ways to the overall instrument
stability thereby adding non-geophysical trending or structure to the combined data records compiled from multiple
missions. These problems can also be exacerbated when comparing instruments with different time histories, for
instance the comparison of two sensors – one during its early-orbit phase with one that has been in space for an
extended length of time. For overlapping spaced-based observations, even with reliable on-board calibration and
degradation correction schemes, time-limited intercomparison campaigns are important to objectively identify
potential systematic errors in one or both instruments. A variety of techniques exist for merging datasets from
different sources – including two different satellites – using statistical models, physical models, and efforts at in situ
calibration (e.g. Chander et al., 2013b; Peterson et al., 1998). Each technique has great strengths and can offer not
just adjustments for merging of datasets, but estimates of uncertainty in long-term stability. Weber et al. (2016)
addressed the issue of requirements on stability for detecting a desired long-term trend from a multiple instrument
time series by accounting for variations in instrumental lifetime and merging biases in a Monte Carlo simulation.
Another approach to addressing satellite uncertainty, based on maintaining traceability through on-board calibration
capabilities using absolute references, has been advocated through the CLARREO and TRUTHS programs
(Wielicki et al., 2013; Fox et al., 2013). For both programs, verification of merging of these new approaches will be
important for validation of expected agreement. For both current as well as proposed satellite systems, the
instrument scientists will decide how to use all available information to make corrections or assign uncertainty
estimates to the data should an offset or drift be detected in overlapping satellite records. Without sufficient overlap
there is a limit to the magnitude of offset or drift that can be detected. This paper presents techniques that can
address the stability of merged data records using observations from overlapping satellite instruments.
Measuring the small changes associated with long-term global climate change from space is both extremely
important and particularly challenging. For example, the satellite instruments must be capable of observing
atmospheric and surface temperature trends as small as $0.1°C$ dec.$^{-1}$, ozone changes as little as $1\%$ dec.$^{-1}$, and
variations in the Sun's output as tiny as $0.1\%$ dec.$^{-1}$ (Ohring et al., 2005). A particular challenge in the design of
climate observing systems is how to preserve data quality and facilitate appropriate evaluations of observations that
extend over a series of missions measuring the same geophysical quantity. A number of in-depth techniques are used
by instrument scientists to understand the fundamental (Level 1) observations, including wavelength scale
corrections, detector responsivity evaluation, and field-of-view sensitivity monitoring. With the regular insertion of
new technology driven by interest in reducing costs and/or improving performance also comes the need to separate
the effects of changes in the Earth system from effects ascribable to changes and gaps in the observing system.
Credible, ongoing programs of sensor calibration and validation, sensor characterization, data continuity, and
strategies for ensuring overlap across successive sensors are thus essential (NRC, 2000a). Multiple efforts describing
key challenges and/or requirements have been published (Chander et al., 2013a; Fröhlich, 2009; Willson and
Hudson, 1991; Willson and Mordvinov, 2003). Adams et al. (2014) revealed up to a 6% relative drift per decade
between different ozone observing satellites, confounding some attempts to detect signs of ozone recovery. Rahpoe
et al. (2015), Hubert (2016) and Tegtmeier et al. (2013) all show that both drifts and biases in current satellite
observing systems are often large compared to the signals of interest. The approaches presented in this paper focus
on developing useful checks on the final data products (Level 2) from multiple instruments.
In the last two decades, there has been an increasing understanding that the merging of records, and the uncertainty
associated with that merging, cannot be considered independently of the final use of the data. NRC (2000a)
highlighted the need for precise inter-satellite calibration, recommending that there should be a 1-year overlap
between successive Ozone Monitoring Profiler Suite missions to allow sensor intercomparison and guarantee long-
term traceability. Analogously, a 1-year overlap in observations of both solar irradiance and spectral solar irradiance
is part of the summary recommendations of Ohring et al. (2005). NRC (2000b) concluded that a special effort is
required to preserve the quality of data acquired with different satellite systems and sensors, so that valid
comparisons can be made over an entire set of observations. Randel and Thompson (2011) explored the utility of
combining the SAGE II ozone observations with tropical measurements from the SHADOZ ozonesonde network, to
study interannual variability and trends. However, not all satellite records have the benefit of such long-term in situ
dataset for intercomparison. Bourassa et al. (2014) quantified interannual variability and decadal trends by
combining stratospheric ozone profile measurements from different satellite systems including using the
Stratospheric Aerosol and Gas Experiment (SAGE) II satellite instrument (1984–2005) with measurements from the
Optical Spectrograph and InfraRed Imager System (OSIRIS) instrument on the Odin satellite (2001–present), noting
significant differences between the different observational sets. These studies indicate that a more robust
understanding of our data records is essential to meeting requirements and making appropriate use of the final data.
Multiple efforts are ongoing internationally to assure that emerging ground-based, in situ and satellite records can be
useful to climate analyses, most notably the Global Space-based Inter-Calibration System (GSICS) which is a joint
effort by WMO and the Coordination Group for Meteorological Satellites (CGMS) to monitor and harmonize data
quality from operational weather and environmental satellites. Harmonized datasets often require adjusting for
offsets, spurious drifts and instrument or location-specific problems (Salby and Callaghan, 1997; Araujo-Pradere et
al., 2011; Dudok de Wit, 2011). To improve the precision and usefulness of multi-instrument time series for
identifying biases, it is necessary to remove offsets between data sources, including those resulting from (a)
calibration differences; (b) spatial and temporal sampling or resolution differences; (c) changes in data processing
versions; (d) inherently different spectral sensitivities; (e) different instrument types with varying inherent vertical
coordinates; and (f) changes in instrument orientation or orbital characteristics or collection times; as examples see
Chander et al. (2013b) and Toohey et al. (2013). These potential problems are further exacerbated by temporal gaps
or insufficient overlap in the satellite records.
In this paper, we estimate the direct impact of length of overlap between satellites to the continuity of data from two
overlapping satellites. We examine three separate factors that are of direct importance to the users and creators of
merged satellite datasets: the quantified offset of the two datasets, the drift between the two datasets, and the impact
of sudden jumps in the data during periods of overlap. We note that intercomparison of satellite records cannot, in
isolation, determine which of two systems is more accurate or stable. Indeed, agreement of two observing systems
can occur when both are similarly inaccurate or similarly drifting and instruments can drift outside of the
intercomparison periods. However, intercomparisons offer valuable, independent assessment useful for developing a
long-term record. For illustrative purposes, we look at ozone, temperature and solar radiation satellite records and
discuss how these three factors can affect the long term records of these parameters. We present techniques for
evaluating overlapping data with the solar dataset because the data are less dependent on satellite drift, diurnal
match-ups and differences in the instrument-dependent field of view. We note that the usefulness of overlapping
data is highly dependent on the length of overlap and the ability to match overlapping data with high precision. In
the final section of this paper, we outline methods for optimizing the set of choices which are needed to create a
long-term and stable climate record under a variety of constraints, most notably economic constraints. Optimization
will result in better use of resources to achieve more accurate and stable merged datasets.
**1.1  Overlap of Earth Observation Satellites**
The value of satellite observations to understanding the variations, climatologies and changes in the Earth's
atmosphere has been profound. Temperature, ozone, water vapor, aerosols and carbon are now understood in ways
unimaginable compared to the pre-satellite era. Continued development of new technologies, including sensor
development, calibration capabilities and refinement of occultation techniques, has resulted in continued
improvements in our observing systems. However, the challenge remains to merge the observations from these
evolving systems into scientifically and societally useful observations.

*Ozone*
Some of the most studied satellite records are the internationally sponsored ozone records. (WMO, 2014 and
reference within, Staehelin et al, 2003). Since 1978, there have been near-continuous space-based observations of
ozone profiles from a combination of missions. Temporal overlaps between these instruments have allowed detailed
intercomparisons to play a key role in assessing the precision, accuracy, and long-term drift of the instruments
(WMO, 1989, 2011a, 2011b; Bodeker, 2001). However, these overlaps have been somewhat serendipitous; little
commitment has been made to ensure the continuity and long-term traceability of the full ground-based, in situ and
satellite ozone measurements. The satellite ozone records, as with the temperature records, benefit from multiple
observations--both satellite and in situ--over the past four decades (Eckert et al., 2014; Staehelin et al., 2003),
resulting in insights into the delivered accuracy and stability of satellite measurements.

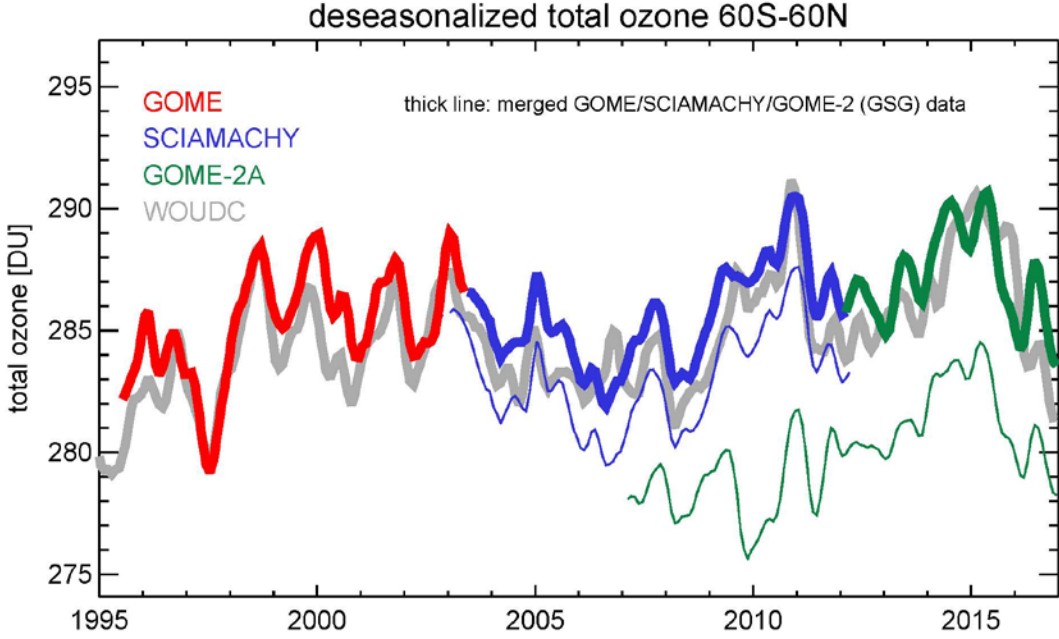

**Figure 1.   The thick and multi colored line is the merged total ozone satellite record (GSG data) from GOME,**
**SCIAMACHY and GOME-2A (Weber et al., 2011, 2016a). The thin blue and green lines indicate the original data before**
**offsets were corrected to create a continuous record. The success of the analytical effort to combine these three satellite**
**records is confirmed by the good agreement of the merged dataset with the zonal mean data derived from ground based**
**Brewer and Dobson data as part of the WOUDC (World Ozone and UV Data Center, update from Fioletov et al., 2002)**
**which is presented as the thick grey line.**
Figure 1 shows the result of merging the GOME, SCIAMACHY and GOME-2A total ozone time series (Weber et
al., 2011, 2015) into a continuous time series. In this case, the SCIAMACHY and GOME-2A observations (thin
blue and green lines) were successively bias adjusted to be continuous with the original GOME data. Biases (offsets)
were determined as a function of latitude in steps of 1 degree using monthly zonal means. Despite extensive pre-
calibration efforts and monitoring of instrument performance, differences are noted between data from the
overlapping satellites. There appears a drop of the original GOME-2 data record during the 2009-2011 period
relative to SCIAMACHY, which seems to be larger than the overall bias between two datasets. In this case the very
large overlap period from 2007 until 2012 was an advantage and no further corrections beyond the latitude
dependent biases were needed to adjust GOME-2. Due to this non-physical drop in the GOME-2A data, the
SCIAMACHY data became the preferred choice in the merged (GSG) dataset during the overlap period (2007-
2011). In contrast the overlap period for SCIAMACHY and GOME was less than 10 months (2002-2003).
Additional corrections beyond a simple bias are difficult and may require the use of external reference data,
although the need for additional corrections may be indicated from satellite overlap data. The long-term stability of
these data are critical for estimating ozone recovery and understanding the complex long-term factors affecting
stratospheric ozone. Lessons from these overlap data serve to offer guidance for future decisions on satellite
observations and overlap periods.
*Temperature*
Perhaps no other set of satellite records has been as studied as the temperature records derived from the MSU and
AMSU satellites. Two distinct challenges complicating the algorithms needed to develop reliable long-term
temperature records are: 1) multiple satellites, in situ and ground based measurements are available each with unique
characteristics and 2) the level of agreement differs with latitude and altitude. Multiple sources of data can
complicate merged data sets because different choices, even when reasonable, can lead to different long-term
characteristics in the record (e.g., Thorne et al., 2005; also see Sec. 3 of this paper). However, multiple records also
allow different groups to produce independent merged data sets which have long overlaps and can be directly
compared. Through these comparisons we gain valuable information about the uncertainties that arise from the
merging process itself, and whether the data sets are stable relative to the requirements of the analysis.

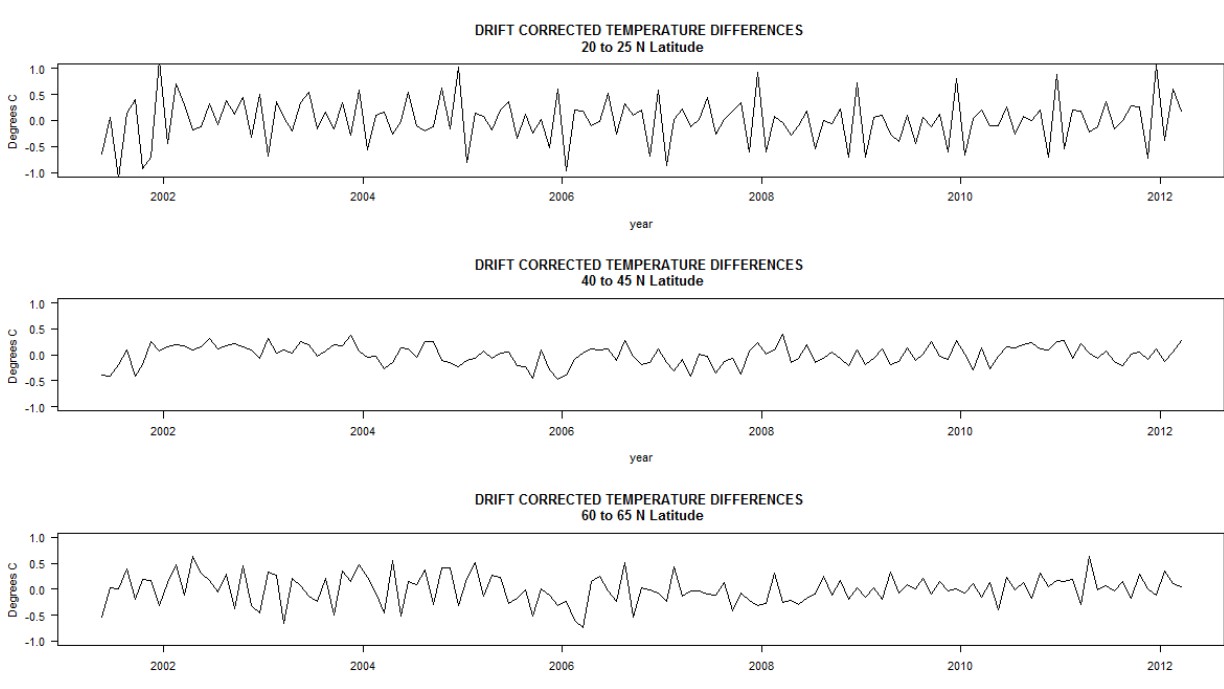

**Figure 2  Monthly differences between a merged MSU Channel 4 –AMSU Channel 9 satellite temperature record and a**
**second merged temperature record based on two satellite-based mid-infrared spectrometers (ACE-FTS and MIPAS), a**
**microwave sounder (SMR), and three satellite-based radio occultation experiments (GRACE, CHAMP, and TSX)**
**(Penckwitt et al., 2015). Shown are the differences once mean offsets and long-term drifts for each latitude band are**
**removed through a statistical regression, in degrees C.  The top plot is for 20-25N; the middle plot is for 40-45N and the**
**bottom plot is for 60-65N. These plots show the variability in overlap is highly dependent on latitude, as is often the case**
**with Earth observations.**
Figure 2 shows the residuals from two independently merged data sets; the results show notable month-to-month
differences between a merged dataset of MSU Channel 4 and AMSU Channel 9 monthly averaged deseasonalized
data compared to a combined dataset from six satellite instruments providing high vertical resolution temperature
data: MIPAS, ACE-FTS, SMR, GRACE, CHAMP and TSX. Both channels are designed to observe the lower
stratosphere. A full description of this merged temperature record and comparisons with the MSU/AMSU merged
record is given in Penckwitt et al. (2015). The comparisons of overlapping data, after offset and drift have been
removed, show several relevant features. First, the variability in the overlapping data varies significantly by latitude.
Second, differences in the merged data sets can be remarkably large -- over a half degree C for latitudinal averages -
- despite differences between data sets being minimized in the merging process and offset and drift between the
compared data set being removed. Third, even when the linear drift over the length of the overlap is removed, the
data show apparent drifts that last for several years in each latitude band. Such variations can limit the usefulness of
the merged records, but can also highlight issues with particular data sources that can then be addressed. The
latitudinal dependence of the variability may indicate regions that are better suited for analysis than others, though in
all cases the physical reasons for the correlated variability and potential drifts needs to be carefully examined. The
stability levels of satellite temperature datasets are critical for understanding the merged and complex feedbacks that
determine regional long-term temperature changes; understanding apparent offsets and drifts between different
sources of information is important, particularly when they are large compared to expected trends.
**1.2 Planning for needed homogeneity**
Detecting and understanding long-term changes require some of the most challenging stability criteria in order for
confidence to be placed on the final results. A number of individuals and coordinated groups have worked to define
the requirements for Earth observations, including the recently completed effort by WMO (2011b), which addresses
the stability needed for various parameters. In the absence of explicit requirements for limits on drift, we suggest
that the standard error of the drift, at the one sigma level, be limited to half of the trend that one is seeking to detect.
For example, if a monitoring system is designed to detect a trend of 0.2 degrees per decade, the unchecked drift of
the system should be less than 0.1 degrees per decade at the one sigma level. While for Earth sciences, the projected
trend is dependent on the climate model and assumptions used to estimate future trends – as well as the location and
time of projected trends--this estimate can be used as a starting point for discussions on how well the drift should be
confined. When the verification of drift cannot be held to the level of projected trends, there can be serious questions
as to the usefulness of the monitoring system for trend identification.
The user communities, including the climate community, continue to request high stability from satellite
observations (WMO, 2011a; Wulfmeyer et al., 2015; Ohring et al., 2005). A variety of ideas have been offered to
allow for more accurate satellite observations: on-board calibration, independent verification, and in-depth modeling
of instrument performance can all assist in characterizing the accuracy, biases, and stability of the satellite
measurements, many based on the fundamental measurement equations and availability of internal instrument
monitoring. There is a long and valuable history of efforts to attempt in-flight calibration, particularly on multi-
spectral sensors (Slater et al., 1996). GSICS coordinates the development of tools to intercompare different Earth
observing systems. (Hewison et al., 2013; Wu et al., 2009) Through careful analysis of spectral signals, relative
stability can be assessed and even small problems with individual sensors can be identified. Ground-based and in
situ observations continue to offer some of the most useful information for constraining offsets and drifts in satellite
instrument as well as providing reliable, direct information on the Earth System. Both campaigns and long-term
monitoring efforts continue to help verify the accuracy and stability of satellite observations. Despite current efforts,
long-term stability and absolute calibration still present a challenge to current internal consistency methods, leading
many to look to other approaches for absolute calibration, traceability, and the ability to verify stability.
Perhaps the most innovative and needed advancements will come though future in-flight calibration approaches. The
development and use of these high accuracy climate benchmark instruments has been advocated for since the early
2000's and described in the NISTIR 7047 (2004) and ASIC3 (2007) workshop reports. These high accuracy
instrument systems will provide two fundamental products of great value to the climate science community: 1)
reliable long-term records of basic climate forcings, response, and feedback for analysis and climate model
verification, and 2) in-flight calibration standards for environmental operational satellite sensors including weather
satellites that do not have a rigorous pre-flight radiometric calibration requirement or the ability to perform
degradation corrections on-orbit. These ideas have been formulated in the visions of CLARREO and TRUTHS, but
may be tested in other reconfigurations (Wielicki et al., 2013; Best et al., 2008; Fox et al., 2013; Tobin et al., 2016).
Until such techniques are developed and tested, ground based and in situ observations continue to offer some of our
best ways of tying satellite observations to traceable standards. Efforts are already underway to estimate
uncertainties due to the matching of independent satellites with these reference sources (Feldman et al., 2011;
Lukashin et al., 2013).
Even with future improvements in satellite observation accuracy, the challenge will remain to understand and merge
records from different satellites – each potentially using its own calibration and collection approaches – to provide a

single observational record. One of the key factors that we can control is the length of overlap between existing and future satellites. Analysis of an overlap record can only give us an estimate of relative drift, but in the absence of traceable in-flight calibration, it is often one of our best checks on long-term stability of the final data products. Understanding that decisions on overlap will directly affect both the cost of monitoring and the value of the final dataset for evaluating long-term changes in climate, we propose approaches to objectively evaluate the length of overlap needed to achieve a specific stability in the merged data record. In Section 8 we offer an approach to evaluate how important overlap is compared to other choices that can help improve a long-term data record.

## 2. Approach

The statistical analysis techniques developed by Weatherhead and collaborators (see for example, Weatherhead et al., 1998; 2000) provide a basis for addressing the length of time needed for adequate overlap based on the magnitude of the signal variance as well as residual noise autocorrelation. In this paper, we perform a case study by applying these techniques to existing SORCE SIM and SOLSTICE instrument data thereby illustrating the use of statistical methodology to estimate the length of overlap needed to achieve records of specified stability. The techniques discussed herein may be useful for instrument scientists pursuing improvements in on-board instrument corrections, but also for mission planning by program managers to ensure the best overlap characteristics of adjoining missions; the basic concepts of uncertainty from merging of datasets are directly useful to those interested in using the data.

In order to appropriately analyze satellite observations, it is necessary to understand and appropriately incorporate the available information on the pre-flight calibration of instruments and in-flight expected behavior. The detailed in-flight circumstances that produce instrument instabilities are highly specific to individual sensors so the best practice is to employ instrument telemetry and on-orbit calibration methods traceable to international standards. Such approaches can be used to develop detailed measurement equations that can account for the occurrence of degradation and correct the measured signal to produce high quality Level 1 data. The measurement equation carries its own uncertainties and, in principal, allows for the estimation of time-dependent uncertainties as a function of mission day. This measurement equation approach is advocated in the "Guide to the expression of uncertainty in measurements" (JCGM, 2008) by the Joint Committee for the Guides in Metrology (JCGM) and relies only on the known and measurable properties of the subsystems that compose the full instrument used for the observation. Instrument teams apply these corrections to produce the final Level 2 data in an effort to provide the most accurate measurements independent of outside data sources. For this study, we assume all relevant corrections have been made to the data to account for known biases and drifts in the instrument.

### 2.1 Introduction of SORCE SIM and SOLSTICE instruments:

For illustrative purposes, we will use two sets of data from the Solar Radiation and Climate Experiment (SORCE) satellite: concurrent data from the Solar Stellar Irradiance Comparison Experiment (SOLSTICE) and the Spectral Irradiance Monitor (SIM). SORCE was launched on 25 January 2003 and has conducted daily measurements of the spectral and total irradiance with only a few gaps in the time series; the longest gap being a 209-day period starting on 31 July 2013. This gap was caused by a reduction in charging capacity of the spacecraft batteries, and has been successfully mitigated by operating the instruments in a day-only operation mode that does not rely on keeping spacecraft subsystems operational on the nighttime portion of the orbit. The instruments for the SORCE mission are described in a series of papers published in *Solar Physics* related to the design, operation, calibration, and performance of the SORCE instruments. Harder et al. (2005a) describes the scientific requirements, design, and operation modes for the instrument. Harder et al. (2005b) discusses the fundamental measurement equations and the pre-flight calibration methodology for the instrument. A third paper (Harder et al., 2010) continues the discussion of the absolute calibration of the instrument describing additional post-launch characterizations using flight spare components and comparisons with the SORCE and UARS SOLSTICE instruments and the ATLAS 3 composite (Thuillier et al., 2004). Additional in-flight comparisons with the ESA ENVISAT SCIAMACHY instrument (European Space Agency, Environmental Satellite, SCanning Imaging Absorption spectroMeter for Atmospheric CHartographY) are discussed in Pagaran et al. (2011). Similarly, McClintock et al. (2005a, 2005b) describe the SOLSTICE instrument design and calibration. Snow et al. (2005) describe the important solar-stellar calibration process that forms the basis of the on-orbit degradation corrections.

**2.2 Set up for SIM/SOLSTICE comparison:**
Solar irradiance is a crucial driver in the Earth's atmospheric system, influencing variability, circulation and long-
term behavior of the atmosphere and having a direct role in atmospheric chemistry for the upper layers of the
atmosphere. A motivation for this solar irradiance study, as is true for most other long-term satellite monitoring
efforts, arises from the need to understand the length of time needed for the overlap of the currently operating
SORCE mission with the next generation Total and Spectral Irradiance Sensor (TSIS). TSIS is currently scheduled
for launch in the fourth quarter of 2017 for deployment on the International Space Station. While Earth observations
often require a minimum of a one year overlap to cover the full range of expected observations, such arbitrary
criteria ignore longer timescale phenomena including ENSO and NAO, and are impractical for covering a full 11-
year solar cycle in a planned overlap period. Here we are applying analytical techniques to understand the length of
time needed to quantify the offset between two satellite observing systems and to understand the drift between two
satellite records (Weatherhead et al., 1998). While it is unclear whether the TSIS/SORCE overlap will mimic the
findings from the comparison of the two SORCE instruments, this effort will examine how potential instrument
anomalies and systematic errors in the degradation corrections affect the ability to determine the length of time
needed to determine a drift difference in the two sensors. For this study, we are using a subset of three years of data
from 18 November 2005 to 31 December 2008 (1140 days), characterizing the time period corresponding to the
descent into the solar minimum condition of Solar Cycle 23 with the minimum value apparently in the January-
February period of 2009. This time period was selected to approximate what would be expected from an overlap
comparison campaign conducted during the descending phase of Cycle 24 projected to be in the 2019 time frame.
However, the Solar Cycle 23 minimum is the longest and quietest time period of the space age (Schrijver et al.,
2011; Araujo-Pradere et al., 2011; Araujo-Pradere et al., 2012), but our analysis does not rely on this situation
persisting into the Solar Cycle minimum. This paper targets common observations of the irradiance in the 280 nm
spectral region which includes the highly variable core of the Magnesium II lines. This region was selected because
the variability of the Mg II lines is an important indicator of solar chromospheric variability and is frequently used
for space weather applications (Viereck et al., 2004; Marchenko and Deland, 2014) and as a proxy for solar
influence on stratospheric ozone and temperature (Hood and Zhou, 1998). It should be noted that the SIM (version
22) and SOLSTICE (version 15) used in this study are used as-reported on the publically available SORCE web
page (http://lasp.colorado.edu/home/sorce/data/ssi-data/). SIM and SOLSTICE corrections are made independently
of one another, but the higher resolution 0.1 nm resolution SOLSTICE data are integrated into a fixed 1-nm bin
centered at 280 nm. The SIM instrument has a FWHM resolution of 1.1 nm with 6 samples per resolution element.
While some offset in irradiance is expected due to spectral sampling used to generate the data products, the
difference is fixed and does not drift as a function time due to the well-defined wavelength scale and spectroscopic
properties of the two instruments (Harder et al., 2010).

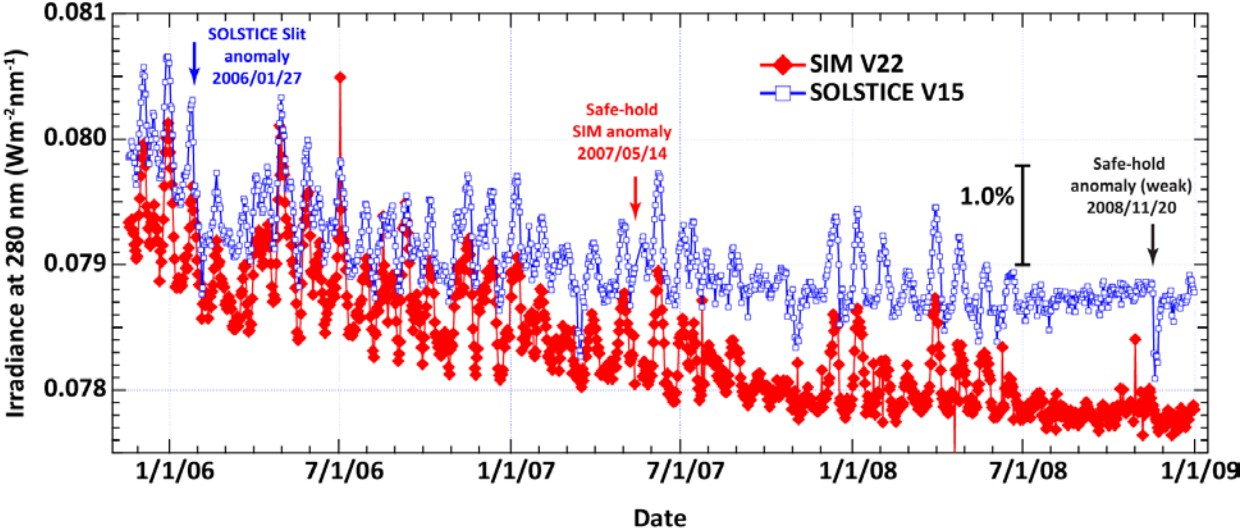

**Figure 3: An 1140-day segment of concurrent SORCE SOLSTICE (version 15) and SIM (version 22) data during the**
**descending phase of Solar Cycle 23. The 27-day variations seen in this plot are caused by solar rotational modulation of**
**active regions dispersed on the Sun and are not due to instrument noise; note that this modulation is apparent in both**
**datasets. This data segment contains all three uncertainty sources identified in this study: offsets, drifts and jumps.**
Figure 3 shows the time series comparison of SOLSTICE version 15 and SIM version 22 used in this study. These
overlapping datasets illustrate three types of inconsistencies that occur in geophysical records, and have been noted
in both the ozone and temperature satellite datasets described above. These three sources of uncertainties combine
and contribute to the length of overlap needed to derive a robust climate record from satellite records.
1. There is an offset of about 0.5% in the pre-flight calibration between the SOLSTICE and SIM. The pre-
flight absolute calibration is on the order of 1-2% so within the ability to absolutely calibrate the
spectrometer. Note that the observed differences in Figure 3 are within the expected pre-flight calibration
uncertainty, but these differences are still large relative to some scientific uses for solar data. The value of
overlapping missions for an appropriate period of time is the ability to verify pre-flight calibration
estimates of uncertainty and potentially improve the long-term datasets for scientific applications.
2. There is an apparent drift in the data between the two instruments. The advancement in SOLSTICE version
15 contains a new correction that removes an annual oscillation in the data induced by a change in the size
of the degradation spot 'burned-in' to the collimator mirror – see McClintock et al. (2005a) for more detail
on the optical configuration. As the Earth-satellite system moves around the elliptical orbit of the Sun a
different illumination occurs on the first optic thereby modulating the intensity of light that propagates
through the rest of the optical system. This same effect occurs in the SIM data but has been a part of the
standard degradation correction for the last versions. SOLSTICE version 15 tends to flatten the apparent
long-term magnitude of the 280 nm variability relative to earlier SOLSTICE versions.
3. There are jumps in the time series related to spacecraft and instrument anomalies. Significant events are
identified in instrument and spacecraft housekeeping telemetry and changes in behavior before and after
these events can be characterized and corrected in the time series. Examples of these phenomena are seen if
Figure 3 where SOLSTICE experienced a failure of the mechanism that changes the entrance slit from the
solar to the stellar mode on 27 January 2006. The slit was moved back into position for continuous solar
observations but did not return to the exact same position so the optical path through the instrument
changed and therefore disrupted the degradation corrections and the wavelength scale. Similarly, a
spacecraft safe-hold event on 14 May 2007 caused the instruments to become very cold and significantly
changed the SIM wavelength scale and perhaps the transmission properties of the instrument. The change
in the SIM wavelength grid is apparent in the uncorrected data, but in Figure 3 the data are interpolated
onto a standard mission-length wavelength scale and does not appear as a jump in this figure. The 2007
safe-hold event had little effect on the performance of the SOLSTICE. The jump associated with the 2006
SOLSTICE slit anomaly has also been corrected and the change in character seen SOLSTICE data at this
time represents the best compromise over the full wavelength range of the instrument.

The next sections of this paper will address the effects of these three anomalies (offsets, drifts, and jumps) in the
SORCE datasets and discuss their impacts on dataset uncertainty. The primary contribution of his paper is to
quantitatively address the impact of the length of measurement overlap on helping verify a specific level of stability
in the final dataset.
**3. Offsets**
Efforts at merging satellite data in the past have focused on deriving offsets to limit relative differences before
combining data from different sensors into a continuous record (e.g. Wentz and Schabel, 1998; Santer et al., 2003;
Smith et al., 2008; Dudok de Wit et al., 2008; Chander et al., 2013b).
One of the most studied issues underscoring the importance of proper treatment of multiple satellite records involves
the corrections and merging of Microwave Sounding Unit temperature records. Christy et al. (1995, 1998, 2000)
accurately pointed out that trends from satellite temperature records were not in agreement with other temperature
records and showed a cooling of the troposphere rather than a warming. Additional work showed that a number of
corrections to the satellite record could make a direct and notable difference on the trend derived from the resulting
data (NRC, 2000a, 2000b; Zou and Qian, 2016). Some of the most salient lessons from this effort were summarized
by Thorne et al. (2005), who concluded, among other points that, "individual adjustments will a priori retain a non-
climatic signal of unknown sign and magnitude regardless of how reasonable and physically plausible the chosen
homogenization approach." The uncertainty of merging satellite data records is a continual challenge with a variety
of approaches employed including comparison to ground-based records, statistical intercomparison of satellites by
latitude, time of day and season, as well as use of physical models to look for appropriate consistencies with
available data. Details of the merging process directly influence the resultant trends and add to the level of
uncertainty in the final datasets (Karl et al., 1986). In this section we consider the case where overlap is non-existent
and for the case where overlap exists we consider the length of time required to achieve a specific uncertainty in an
overlapping set of data. These cases illustrate the need for overlap periods of sufficient duration to make a
quantifiable improvement in the long-term record.
We consider the straight-forward method for merging two sequential (non-overlapping) data records by requiring
the mean level of the three years of data prior to the discontinuity be equal to the mean level of the three years of
data after the discontinuity. In such a situation, those six years of data are being forced, by the algorithm, to have
very little trend. Imagine a situation where there are two such discontinuities in a twenty year record; more than half
of the data has been coerced to have virtually no trend, making the resultant data unreliable for many long-term
monitoring uses. The case of no overlap can occur due to a variety of reasons, including the sudden loss of a
satellite, or problems on launch of newer satellites. The end result of any offset correction will have a direct impact
on the magnitude of the resulting trends.
To estimate the time needed for overlap requires an estimate of what the overlap time series would look like. We use
monthly averages, a common standard in many climate-related research efforts for several reasons: monthly
averages avoid the match-up issues and potential non-linearity of short-term features, such as storms (Araujo-
Pradere, 2004), and offer enough resolution to observe long-term behavior of the match-up.[1] If, as is the case of the
SOLSTICE-SIM overlap, the difference between the two datasets look like Fig. 4 with significant agreement in
observed variability, the paired data can be used to estimate offsets and uncertainties in the derived offsets.

---

[1] While it could be argued that there is nothing unique about the time-step of one month, it is a common practice in climate analyses. However, the example datasets used in this paper measure extra-terrestrial solar radiation. With the Sun's rotation of 27.2753 days, we have a natural timeframe close to a monthly average. In Appendix A we carry out the calculations in this paper with monthly averages and with averages based on the solar rotation schedule; we see no notable change in the basic conclusions adopting the more natural solar rotation schedule instead of monthly averages.

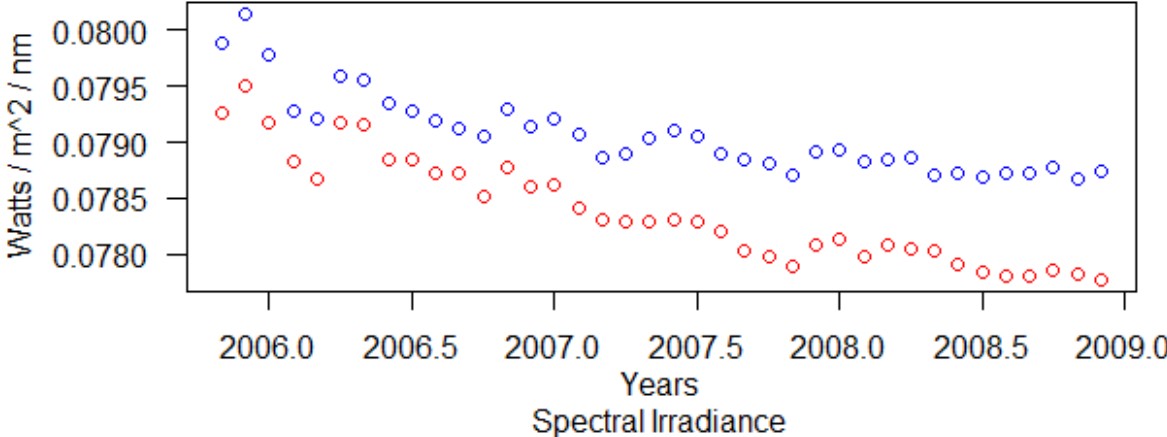

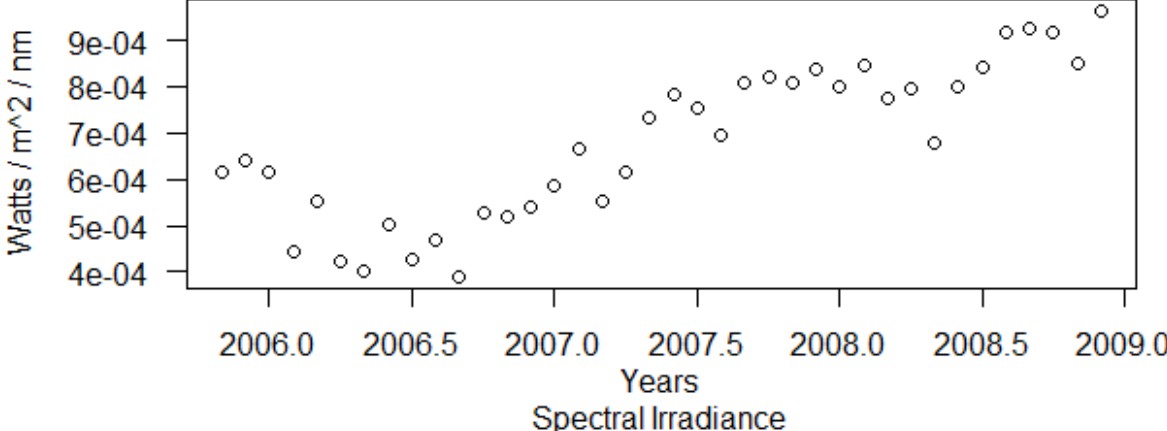

**Figure 4: Monthly averaged SOLSTICE data and SIM data (top plot) and SOLSTICE-SIM (280 nm Mg II) data (bottom plot) (watts m$^{-2}$ nm$^{-1}$) as a time series for the period October 2004 through December 2008. The data show that the observed differences are small, but do not appear to be stable, thus a simple level shift to bring the two datasets into agreement may not fully address the match-up and stability issues of the dataset.**

The overlap data depicted in Fig. 4 show a mean difference between the two datasets of $6.8*10^{-4}$ watts m$^{-2}$ nm$^{-1}$, with a standard error on this mean of $2.7*10^{-5}$ watts m$^{-2}$ nm$^{-1}$ when the classic standard error calculation ignores autocorrelation. However, this figure does not support the assumption that the observed differences between SOLSTICE and SIM are stable and would continue beyond the observed end of the analysis period of December 2008 because of the apparent drift in the differences. For cases when a drift is not involved, we can make use of the standard formula for the standard error on the mean of the observed time series of differences when simple autocorrelation is present:

$$SE_{mean} \cong \sigma \Big/ \sqrt{n} \ \frac{\sqrt{1+\varphi}}{\sqrt{1-\varphi}} \tag{1}$$

Where $\sigma$ is the observed magnitude of variability of the observed differences in monthly averages; $\varphi$ is the observed
autocorrelation in those differences, and n is the number of months of observed overlap. This estimate of Standard
Error of the mean is dependent on the data behaving as an autoregressive with time-lag of one month, AR(1), with
the underlying interventions behaving approximately as a Gaussian distribution. This more appropriate formula
gives a standard error on the mean of $5.2*10^{-5}$ watts m$^{-2}$ nm$^{-1}$, notably larger than if autocorrelation is ignored.
Monthly averages have a broad range of uses in environmental sciences for trend detection, development of
climatologies and monitoring the behavior of the Earth. Monthly averages can remove higher frequency noise and
some sampling match-up problems, but they can also obscure important details and can often introduce their own
biases, especially when sampling is irregular in time or space (Toohey et al., 2013; Toohey and von Clarmann,
2013). We show the behavior of the underlying interventions as Gaussian and our tests for AR(1) in Appendix B.
We can invert the formula for the Standard Error on the mean in Eq. (1), and solve for *n* resulting in the time to
estimate the mean offset between two satellites for a given accuracy as:
$$Months\ to\ Estimate\ an\ Offset \cong 1.96^2 \ \sigma^2 \Big/ Offset\ Limit^2 \ \frac{1+\varphi}{1-\varphi} \tag{2}$$

The above formula shows that for a given magnitude of variability and autocorrelation in monthly satellite overlap
data ($\sigma$ and $\varphi$ respectively); the length of overlap needed is inversely proportional to the square of the accuracy
desired for the offset estimate. The factor of 1.96 is to support a 95% confident limit on the offset with a 50%
likelihood of detection; if more confidence is needed in the offset, a higher factor can be used based on classic
statistical tables. Thus, if we can identify the level of uncertainty we can accept in a merged record due to the
overlap offset (SE$_{mean}$), and if we have some understanding of the behavior of overlap differences ($\sigma$ and $\varphi$), either
from advance estimates or from early analysis of offset data, we can appropriately identify the length of overlap
needed in a manner that is respectful of the inherent cost of added months of satellite overlap. If a higher level of
certainty than 95% is required, the 1.96 factor is adjusted appropriately according to normal distribution tables. For
small number of months, the 1.96 will need to be adjusted for the student-t distribution which allows a larger
uncertainty when a small number of points are used. With the example used in this paper and shown in Fig. 4, we
observe a magnitude of variability, $\sigma$, of $1.7*10^{-4}$ watts m$^{-2}$ nm$^{-1}$ and autocorrelation, $\varphi$ of 0.89. If we want an Offset
Limit of 0.0008 watts m$^{-2}$ nm$^{-1}$ (which is one percent of the mean of SOLSTICE during the overlap period), then the
number of months would need to be 5 months using the student-t distribution which offers 2.8 as the appropriate
factor in place of 1.96. Note that to achieve the 95% confidence limit, we must use the appropriate student-t
distribution, or approximately 1.96 multiplier in the large number limit, to assure we have the desired confidence on
our overlap adjustment. Note also that this is a recursive effort because the answer, number of months, is a function
of the multiplier, which is itself a function of the number of months. This exercise is not overly onerous, because the
formula offers an estimate of length of time needed to limit uncertainty in an offset, and such an estimate is rarely
precise to many significant digits. We conclude for the datasets we have been exposed to that after roughly two
years of data collection the large number limit of 1.96, may be considered appropriate.
The impact of the offset on the use of the data is critically important to the final analysis. While a "best" merged
dataset may be produced from multiple satellites, users should never ignore the added uncertainty due to merged
data sources. Using the merged data without including the impacts of the merging would result in smaller standard
errors in computing means, variability in trends, than is actually appropriate. The magnitude of the impact of the
offset correction is dependent on the use of the data. Two cases are considered here for illustrative purposes. If the
merged dataset will be used to estimate the impact of storms on a stable electrical grid, and the impacts have been
estimated from the effects observed using the first satellite record, an uncertainty of 0.2% means that the new solar
storms may well be off by $\pm0.2\%$ and the uncertainty in impacts need to be appropriately calculated and conveyed.
If the merged datasets will be used to estimate long-term trends, then the impact of an uncertainty of $\pm0.2\%$ means
that any trends derived will be affected by that level of uncertainty carried out through the length of dataset used for
analysis, and may affect the significance of the expected trend, if care is not taken to reduce the uncertainty in the
overlap adjustments.
**4. Drifts**
While offsets are routinely addressed in the merging of satellite datasets, potential drifts in satellite data are also
critically important to many of the final applications of climate data, most notably trend detection both of the direct
parameter being observed and observations that are dependent on the observed parameter. There are several
fundamental factors that can contribute to a drift in satellite observations including decay of instrumentation and
changes in satellite orbit. Efforts are ongoing to minimize the impact of these factors, but all corrections involve
assumptions and each satellite may invoke different approaches to monitor and address stability. The merging of
satellite records, at a minimum, needs to test for potential drift between the overlapping satellite records. The
amount of drift that can be detected through satellite overlap depends on the length of overlap period and on the
quality of the match-up in overlapping data.
The impacts of undetected drift will have direct impact on the scientific results derived from the data. Bourassa et al.
(2014) showed that the uncertainty in drift from a continuous record from multiple satellites is critical to long-term
monitoring of the Earth. Rahpoe et al. (2015) find intercomparisons of six different ozone limb measurements to
drift relative to each other at a statistically significant rate, sometimes as high as 5% per decade or more. However,
most drifts were statistically insignificant due to the limited length of data records--generally less than 10 years. In
the case of solar viewing instruments, BenMoussa et al. (2013) discuss in detail causes and effects of degradation in
a variety of different instruments that span nearly two decades and cover a broad wavelength range. They conclude
that there is no single best method to correct and monitor degradation and the correction schemes for overlapping
missions are likely to be very different depending on the instrument hardware selection. An important example of
this is well documented in the efforts to correct drift in the CIRES instrument (Cloud and Earth's Radiant Energy
System, see Loeb et al., 2016 and references therein). In this report, long-term stability was linked to loss of optical
transmission due to UV exposure and molecular contamination, very similar to the mechanisms discussed in
BenMoussa et al. (2013). Fruit et al. (2002) have addressed the effects of energetic particle on glass transmission,
but inhibiting and characterizing carbonization of optical surfaces remains a steadfast and unsolved problem. In each
case, evaluation of how best to characterize the drift takes place. For the Solstice-SIM data overlap, we noted that
the differences between the two sets of data showed lower variability than the ratio of the data, indicating an offset
would be better modeled as an additive adjustment. In many cases, uncertainty on satellite records' drift can be even
more significant to scientific uses of the data than the offsets from one instrument to the next. In this paper, we look
to see, to what extent, some confinement of the problem may be achieved through appropriate overlap of
independent instruments.
If we can quantify the level of drift we would like to be able to detect, and if we can estimate the level of variability
in the overlapping data, using approaches from Weatherhead et al. (1998), we can estimate the length of time
necessary to observe a drift of that magnitude in an overlapping dataset. Weatherhead et al. (1998, 2000) have
shown that one can estimate the length of time to detect trends in environmental observations. This approach is
applied to estimate the time to measure a differential drift with a specified uncertainty in the observations taken by
two different systems. When detection is considered at the 95% confidence level, estimated overlap for detection is:
$$Months\ to\ Estimate\ a\ Drift\ \cong\ 12*\left[1.96\ \frac{\sigma}{|drift|}*\sqrt{\frac{1+\varphi}{1-\varphi}}\right]^{2/3} \qquad\qquad (3)$$
Where |drift| is the absolute value of the magnitude of the differential drift, $\sigma$ and $\varphi$ are the magnitude of variability
and autocorrelation, respectively, of the differenced monthly data once any existing trend is removed. We can
identify the drift we would like to have the capability of detecting; and estimate both $\sigma$ and $\varphi$ from existing data –
either from observations or modeled experiments. As an example, a drift of 0.1% per year of the observed SOLSTIC
data, (0.00008 watts $m^{-2}$ $nm^{-1}$ $yr^{-1}$), with the observed variability in the overlap data (sigma of $8.59*10^{-5}$ watts $m^{-2}$
$nm^{-1}$ and phi of 0.57) results in 2.5 years likely needed to achieve that uncertainty.
It may be noted that the natural world has variability ($\sigma$) and autocorrelation ($\varphi$) that are inherent, and may change
slightly over time. A distinct advantage of satellite observations can be the frequency of the observations.
MacDonald (2005) has shown that the monitoring approach can have a direct impact on these values, as well:
monitoring less frequently – perhaps only once or twice a month – results in higher variability and slightly lower
autocorrelation in our dataset. As encapsulated in Equation1, an increase in the number of measurements per month
improves the detectability (shortens the number of years to detect a given trend), but only up to the limit of the
system's natural variability. For each situation of overlapping of satellite missions, the results will depend on the
method of observation and the parameter being observed; for Earth observations, the results can also depend on
location and even time of year.

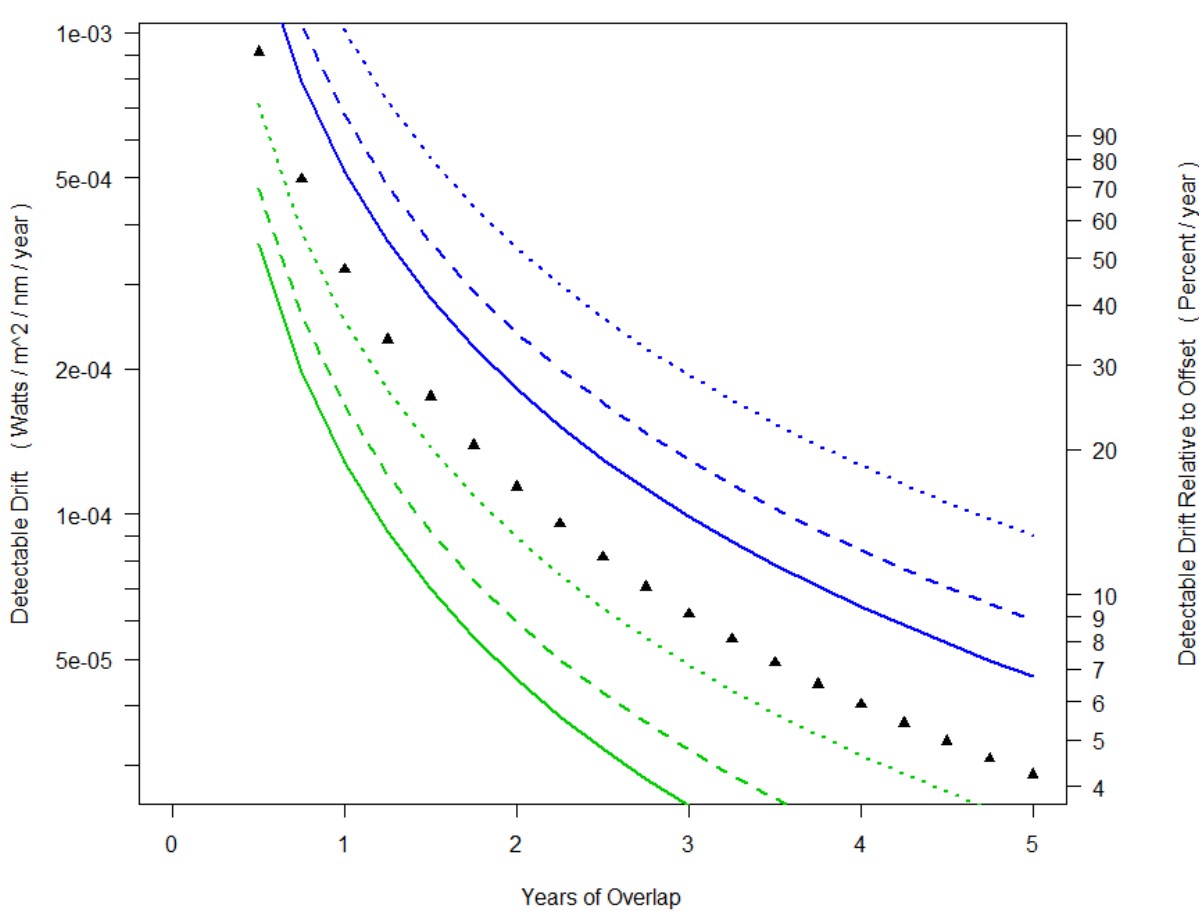

**Figure 5: The ability to constrain a detectable drift is a direct function of the number of years of overlap and the quality**
**of the overlap data. Black triangles offer the detectable drift in the difference between the SOLSTICE and SIM satellite**
**instruments as a function of the overlap period; calculations made use of the observed characteristics of the Solstice-SIM**
**monthly differenced data: magnitude of variability of $8.58 * 10^{-5}$ watts $m^2$ $nm^{-1}$ and autocorrelation of 0.57. Green lines**
**assume a smaller magnitude of variability in the overlap data (half of what is observed in SOLSTICE-SIM) and**
**autocorrelation of 0.4 (solid), 0.6 (dashed) and 0.8 (dotted). Blue lines assume a larger magnitude of variability in the**
**overlap data (twice what is observed in SOLSTICE-SIM) and autocorrelation 0.4 (solid), 0.6 (dashed) and 0.8 (dotted).**
Using the SOLSTICE-SIM data as an example, tremendous accuracy gains are achieved for each year of monitoring
for the first few years of overlap, with diminished returns after that. Respecting the cost of overlap, and making
appropriate calculations with emerging overlap data, an appropriate overlap plan can be estimated to allow for
scientific standards to be met. Figure 5 shows that for a given magnitude of variability and autocorrelation observed
in overlap differences, we calculate the number of years of overlap needed to detect a specific level of drift. In this
case, using the SOLSTICE-SIM data as an example, the magnitude of detectable linear drift drops from $1.1*10^{-4}$ to
$0.6*10^{-4}$ watts $m^{-2}$ $nm^{-1}$ $year^{-1}$ by allowing the overlap to be three years, instead of two years. The level of agreement
of data from the two instruments results in the magnitude of variability and autocorrelation observed in the
differences. For observations that are very much in agreement with each other, we can expect low variability, and
thus, a relative drift can be detected earlier, as represented by the green lines in Figure 5. For a poor match between
the overlapping observations, we can expect longer times of overlap will be needed as represented by the blue lines
in Figure 5. The relative drift on the right side of Figure 5 offers information on the percentage basis of the overlap
data, not in the raw SOLSTIC or SIM datasets, because it is the fundamental behavior of the differences that
determines the information content of the overlap period. It is important to note that drifts in overlap data, as drifts
in nature, can be approximated as linear, logarithmic, or a variety of other representations, as the data and the
physics of the situation suggest; for Fig. 5, we assume an approximately linear drift over the time period of the
overlap. For satellite observations, a number of changes are expected over the lifecycle of the instruments; all known
and expected changes are approximated and adjusted based on current best understanding. However, particularly
with new technologies, these assumptions must be checked by careful evaluation of the data, thus emphasizing the
importance of an adequate overlap period to help confine potential drifts to a specified level. While pre-launch
calibration may indicate drift will be less than a specific level, the ability to verify this will depend on independent
intercomparisons of observations.
Although no error bars are offered in Fig. 5, it is important to remark that when estimating how long it will take to
detect a specified drift, two statistical levels must be considered: one that identifies the meaning of "detecting a
drift" and a second that identifies the likelihood of detection of that drift in the specified period of time, if that level
of drift is the true, long-term drift in the overlap. Of course, it is possible to detect smaller drifts than the value
obtained from any particular point (drift-overlap pair) of this figure, as it is also possible to determine a given drift a
few months earlier or later than the value obtained from the point in the figure. Comments on the appropriate
interpretation of the likelihood of drift detection are discussed in detail in Appendix C, where two non-standard,
dimensional error bars are introduced in the figure to help the reader to understand this uncertainty.
**5. Jumps**
Jumps are permanent or semi-permanent level shifts in the data that occur at specific points in time and are not
attributable to the parameter being observed; jumps could represent a change in sensitivity of an instrument or a
change in location or orientation of the satellite. To ignore such events results in greater uncertainty in the
appropriate offsets and in artificial drifts in the final data set. If the magnitude and time of the shift are known, then
the data can be adjusted before being analyzed for offsets and drifts. If the magnitude and timing of jumps are
unknown, as is more often the case, and must be derived from the data, the presence of such level shifts increase the
uncertainty of both the offset and drift in the overlapping data and hence lengthens the time necessary to achieve a
high quality final dataset. Continuous satellite overlap can make the ability to identify and understand sudden jumps
significantly easier, but this is often beyond current monitoring approaches.
It is tempting to believe that jumps can be easily identified and corrected. Testing this belief, Free et al. (2002)
carried out a comparison of seven different groups examining radiosonde records with the intent of identifying,
quantifying and correcting observed discontinuities in temperature sonde time series. The different groups identified
a widely different number of discontinuities using a range of techniques. They further differed significantly on the
magnitude of corrections and even, at times, the sign of the needed correction, resulting in changes to observed
trends by between 35% and 80%. Thus, even identified and corrected discontinuities introduce some uncertainty in
the long-term stability of the record. Weatherhead et al. (1998) showed that these corrections can be quantified with
advanced statistical techniques. The resulting uncertainty in long-term stability increases the number of years
necessary to detect trends, independent of how large the correction is. Hurrell and Trenberth (1997) point to the
importance of two small, discrete, downward jumps in merged satellite records that dominated the trend results for
tropospheric temperature records.
Jumps can occur for a variety of reasons related to instrument changes (e.g. Brown, 2013). Depending on the
physical source of the jumps, the effects can last from less than a few hours to multiple years. In some cases, true
jumps occur in the parameter being observed. In many other situations, the observing system or assumptions used in
the algorithms are responsible for the jumps and there is a desire to identify and remove these spurious jumps. The
statistical removal of a spurious jump involves two steps: identifying a jump in the overlap differenced data and
estimating the magnitude of the needed correction. A variety of approaches are used to both identify jumps (Jaxk et
al., 2007; Vincent et al., 1998, 2002; Ducre'-Robitaille et al., 2003) and correct for these jumps in satellite and non-
satellite observing systems (Karl and Williams, 1987; Mitchell and Jones, 2005). When information is available to
identify the timing of a jump, there is considerably more confidence in the correction for the jump because physical
interpretation is easier and therefore corrections can be physically based rather than statistically based. If there is no
external information on timing, then one has to consider that there could be other jumps below the threshold for
detection and estimates for how the instrument is behaving are more uncertain. For jumps in overlapping datasets,
the correction brings into question the magnitude of any derived offset as well as the magnitude of any drift in the
overlap period. For jumps in observing systems when there is no overlap, the challenge of appropriately identifying
and correcting jumps is notably more difficult, again pointing to the value of redundant observing systems when
possible. We focus on the ability to detect and understand the jumps that last more than a few months, as they may
be the interruptions that can cause the most serious damage to long-term records, particularly those used in the
context of climate research. We consider the two cases separately of how a jump affects the offset estimate and how
a jump affects the drift estimate.
While ancillary data about the instrument or the observed parameter may be used to identify the existence and
timing of a jump, deriving the magnitude of correction by examining the data is dependent on the amount of
variability (both magnitude and autocorrelation) in the overlap difference data. For the case of an otherwise stable
offset between the two sets of satellite data, the added uncertainty on the true offset is enhanced based on the
uncertainty of the jump. The impact of the jump on offset estimates is a minimum when the offset occurs in the
middle of the overlap period, because maximum information is available to identify the size of the jump. For the
case of a drifting offset between the two sets of satellite data, the added uncertainty on the true drift is also enhanced
based on the timing of the jump. Because the impact of the jump is co-linear with the derived drift, estimating the
overall drift in instrument offset is more difficult in the presence of jumps. If we assume that we are going to fit the
environmental data to a linear statistical model of the form:
$Environmental\ Data = Mean + Linear\ Drift + Offset_\tau + Noise(0,1)$                    (4)
where the Mean, Linear Drift and Offset at time $\tau$ are derived from the data. The residuals ($Noise(0,1)$) are modeled
as an AR(1) process with mean=0, and both $\sigma$ and $\varphi$ derived from the observations. With such a statistical model,
Weatherhead et al. (1998) show that the impact of the additional jump term on deriving an accurate estimate of drift
is dependent on the timing of the jump.
$Months\ to\ Estimate\ a\ Drift \cong 12 * \left[1.96\,\frac{\sigma}{|drift|} * \sqrt{\frac{1+\varphi}{1-\varphi}}\,\right]^{2/3}\,\frac{1}{[1-3\tau(1-\tau)]^{1/3}}$                    (5)
Where $\tau$ is the fraction of the data before the identified jump occurs. Thus, a longer time of overlap is required for
accurately confining a drift in overlap data: As opposed to the impact of jumps on estimating offsets, the impact of
jumps on derived drifts is largest when the jump occurs in the middle of the overlap period ($\tau$=0.5), where the
amount of time is increased by a factor of 1.59. This increase is in length of time needed to estimate a drift is due to
the similar temporal signature of both drifts and jumps on a long-term record as illustrated in Fig. 6. Equation 4
assumes that any drift and offset will be fitted to the data simultaneously. If data will be fitted sequentially to an
offset and then to a drift, the derived drift will be considerably smaller than if the data were fitted to a drift and then
to an offset because the two functions (drift and offset) are not orthogonal and thus the derived results for magnitude
of offset and magnitude of drift are not commutative. To be explicitly clear, correction of offsets in advance of
deriving drifts can artificially minimize the amount of observed drift, while ignoring offsets (perhaps because they
are not easily detectable) can either add to or diminish the derived drift.

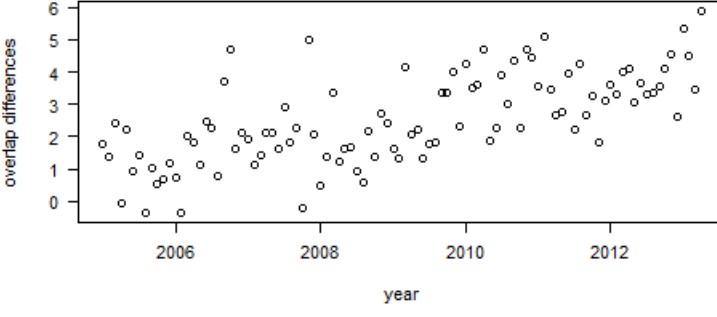

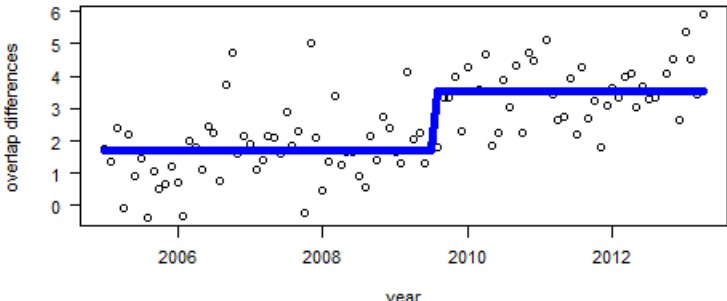

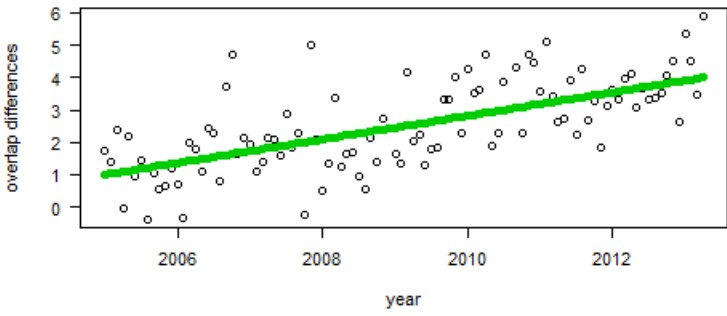

**Figure 6: This dataset was created to visually show the potential impact of a spurious jump on the estimate of offsets and**
**drifts. In all three plots, the same data are shown, with the second plot showing how the data could be modeled as an**
**offset. The data were actually created by adding a linear drift to simulated autoregressive data, as shown in the third plot.**
**The confounding nature of jumps and drifts cannot easily be separated; although ancillary data can be extremely helpful.**
**Note these are synthetic data with arbitrary units.**
In the case of the SORCE instruments, these jumps are mostly prompted by changes in the performance of hardware
subsystems of the spectrometers. For example, a significant jump occurred in the SIM on 14 May 2007 related to the
instrument becoming very cold during a safe-hold event, and upon recovery, the instrument did not return to the
same state as before the safe-hold. This event produced a significant change in the hardware that controls the
wavelength drive (see Harder et al., 2005a for a description of the wavelength drive mechanism). Even after the
wavelength correction a residual jump in irradiance level was still observed; the most likely cause of this jump is
due to a change in light transmission caused by a change in optical path through the instrument that is different than
before the safe-hold event.
For the particular case of Fig. 3, the jump appears to be adequately corrected, but other wavelengths show a
discernable discontinuity. A similar observation can be made about the SOLSTICE slit anomaly in 2006 (see Fig. 3).
Most disconcerting about these events is the possibility that the jumps can be disguised as a change in stability and
produce results similar to what appears in the discussion of Fig. 6.
**6. Impacts of Uncertainty in Drifts**
The primary purpose of overlap in instruments is to understand how two different instruments are responding to the
parameter each is intended to measure. Uncertainty in the long-term stability of data products directly affects the
level of confidence one can have in the long-term, merged datasets. Logan et al. (2012) noted that differences as
small as a few ppb in ozone records with 2–3 years overlap lead to different trends for 1995–2008. Frith et al. (2014)
use Monte Carlo techniques to estimate the uncertainty of the SBUV MOD merged data set based on both overlap
information and validation against independent data sets.
The impact of undetected drifts in overlapping data sets on the merged record can be illustrated with a simple Monte
Carlo representation, as shown in Fig. 7. Consider two satellites launched with seven year lifetimes. The second
satellite (blue trace) overlaps the first satellite (green trace) by either one or two years (top plot and bottom plot,
respectively). Each satellite launches with an uncertainty estimate on the pre-flight absolute calibration of its
measurements and some level of unidentified drift in the instrumental record, acknowledging that all identified
drifting factors have already been corrected to the best abilities of the instrument team. For this illustration, we
assume our simulated instruments have the same variability and autocorrelation during their overlap as the
SOLSTICE-SIM data used in the previous sections. Accordingly, an overlap of one year implies a drift in the
differences smaller than $3.28e^{-4}$ watts $m^{-2}$ $nm^{-1}$ $year^{-1}$ (1 standard deviation; roughly 48 % $yr^{-1}$) would be
undetectable, and a relative drift of $1.16e^{-4}$ watts $m^{-2}$ $nm^{-1}$ $year^{-1}$ (1 standard deviation; roughly 17 % $yr^{-1}$) would be
undetectable with two years overlap (see Figure 5). For illustrative purposes we show 100 simulations generated
based on these potential relative drift values. In addition we depict a scenario in which the pre-launch absolute
calibration of the second instrument is better constrained, which can occur when advanced technologies are
introduced. Such a priori information about the individual data sets is important; in this case a user might choose to
adjust the first satellite record to the second to take advantage of the smaller absolute uncertainty in the second
instrument.

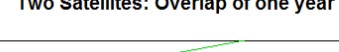

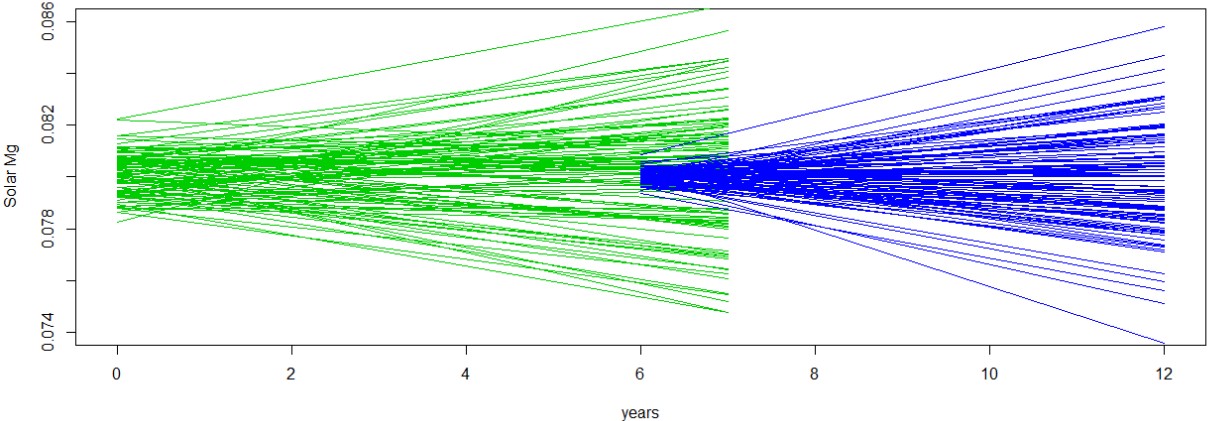

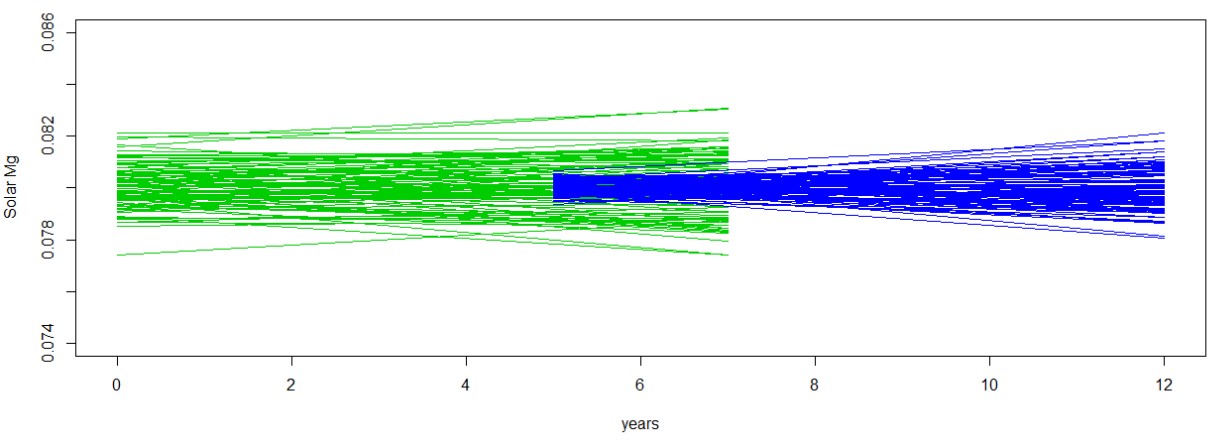

**Figure 7: Two overlapping satellite observations (shown in green and blue) simulating 280 nm Mg II irradiance (watts m$^{-2}$ nm$^{-1}$) with an expectation for some absolute pre-flight calibration uncertainty and some unidentified on-orbit drift. The amount of unidentified drift can be estimated in the overlap period, resulting in the possibility of a correction to the data. The uncertainty in the estimate of the drift is notably smaller with a two year overlap period in satellites compared to a one year overlap.**

We note that the drift uncertainty depicted here only represents the uncertainty associated with adjusting one data set to the other; it does not include the absolute drift. If, as an example, we assume our two simulated instruments had the same relative drift as observed in the SOLTICE-SIM example (~1.67e$^{-4}$ watts m$^{-2}$ nm$^{-1}$ year$^{-1}$, see Figure 4), this drift is not detectable with one year of overlap, but is detectable with two years overlap. However no amount of overlap can reduce the true absolute drift of either satellite dataset. As with any instrument intercomparison, satellite overlap by itself cannot replace calibration, but it does offer valuable information that significantly complements all other available information. However, this illustration can help guide how large an impact any confinement of potential drift may be and can offer insights into whether uncertainty on the level of drift is too large for the intended use of the data. Based on this type of analysis, some have argued for the value of redundancy in observing systems (Weber et al. 2016b) and the value of continued replacement of satellites to minimize the extent of unchecked drift in a single record (Stolarski and Frith, 2006, Frith et al., 2014 and Weber et al., 2016a) to potentially reduce the absolute uncertainty in merged records.

**7. Application to In Situ and Ground Based Observations**
Both in situ and ground-based Earth Observations often require targeted efforts and planning to assure a continuous
record based on discontinuous observing systems. In some cases, overlap is impossible when changes take place; for
instance when an expensive instrument moves location within a region or local changes, such as urban development,
directly affect the local observations. However, in some cases, the approaches presented in this paper can have direct
relevance. WMO's Global Climate Observing System's Reference Upper Air Network is currently evaluating the
transition from one sonde instrument package to another. Such a change in instrumentation is likely to affect the data
stream particularly when extreme values are concerned. The planned overlap period and detailed planning of the
intercomparison of results is currently underway within the GRUAN leadership team. In a similar situations, UK
Met Office evaluated the impact on a the switch from RS92 sondes to RS41 sondes, following a careful study
carried out jointly with Vaisala and the UK Met Office making use of simultaneous launches of 30 sets of two RS92
and two RS41 launches. The ability to compare the data simultaneously, using the same balloon system and doing
the intercomparisons with synchronized GPS times allowed for much smaller differences in the statistical limit to be
placed on the impact of transition from the RS92 to RS41 observations. This controlled study is the ideal situation
and contrasts with no overlap as noted by Weatherhead et al. (1997) which showed the large impact of sudden shifts
of radiation measurements, some of which were due to shadowing of the instruments by newly built structures. The
sudden shifts without overlap rendered the data inappropriate for trend analysis because the uncertainty due to the
level shifts were too large to allow detection of trends expected from ozone loss.
**8. Optimization and economic benefit**
Decisions to improve a single observing system can rarely be made without consideration of the potential impact on
the support of other approaches to improve monitoring. Figure 5 shows that drift detection accuracy improves as the
number of overlap years increases. Improvements in drift detection capability decrease as the number of overlap
years increases, but the optimal overlap duration is difficult to identify unless restrictions, such as cost, are
considered. If a specific stability criterion is the objective, the minimum required overlap can be directly determined
to meet that criterion. Assuming that the total costs increase as overlap time increases (this value can often be in
excess of $1-2M per year), from an economic perspective the minimum required overlap time should also be chosen
as the most cost-effective. Although technical criteria are critical to understanding optimal data series overlap,
decisions in the policy arena are often based on economic analysis and thus it is important to better understand how
economic analysis frames the allocations of resources. "Optimal" in such discussions is structured on criteria
measured in societal benefits and costs.
In addition to overlap time, a number of choices may be proposed to improve observational climate records
including, as already mentioned, improved pre-calibration of satellite systems, intra-satellite calibration
mechanisms, redundancy in observing systems, and campaign verification of observations. In considering any
proposed set of approaches to achieve a specific stability criterion, the least cost option or combination of options
should be chosen. However stability criteria are often not available or substantiated. Under the constraint of fixed
budgets, and without specific criteria, the maximum overlap period affordable would be optimal. If there are
multiple approaches being considered, the combination of improvement approaches to achieve the greatest stability
in the data within the exogenously determined budget would be optimal. Given the complementary information in
various approaches to improve both absolute and relative calibration, linear optimization approaches may need to be
developed to identify the best mix to achieve optimal calibration.
Without specific stability criteria or budget constraints, economic criteria suggest choosing the overlap period or
combination of approaches to achieving data stability that provides the maximum net societal benefit (i.e., total
benefits minus total costs).[2] Identifying the societally optimal choice implies choosing the overlap (or possible mix
of calibration methods) where marginal cost equals marginal benefits. Marginal costs are the change in program
costs to achieve one unit of improvement (e.g. watts $m^{-2}$ $nm^{-1}$ $year^{-1}$ in detectable drift as per Fig. 5). From an

---

[2] Appendix D provides a more technical explanation of the optimal choices under difference decision situations from
an economic perspective.

economic perspective, the marginal benefits of one unit of improvement in the data quality should be measured in
terms of potential changes in societal outcomes from the use of improved information (a value of information or
VOI approach (Laxminarayan R. and M.K. Macauley, eds. 2012)). While there has been some work on the value of
information from satellites (Donaldson and Storeygard, 2016; Cooke et al., 2014; Macauley, 2006), there have not
been many applications of economic analysis to determine optimal observational systems. Morss et al. (2005)
provide an overview of relevant economic concepts and theory for optimal design of observational systems based on
benefit cost tradeoffs.
Given the limited number of applied studies on societal benefits of satellite data, standard approaches would need to
be adapted to further understand the value of stable records and develop the decision making tools to optimize
observation systems. Weatherhead et al. (2015) have gathered community input to help identify key science
questions that need to be addressed, while Feldman et al. (2015) have outlined some tools for assuring observing
systems can meet these approaches. The general field of climate observing system simulation experiments C-OSSEs
is gaining serious attention from both scientists and science managers internationally.
This paper provides guidelines for what may be considered idealized situations. Approximations have been made
about the linear nature of drifts, and timescales of jumps that are likely approximations to considerably more
complicated instrumental response. Individual judgment is needed to apply the results from this paper as instrument
characteristics often change over time and PIs will often have additional information that will guide their decisions
about the quality and stability of instruments. Ground-based instruments would likely add further information to
help evaluate stability. Notably, the beginning and end of most satellite missions are the periods where most
challenges occur in the instruments and may alter the guidelines presented here. As an example, instruments
behaving very badly are perhaps not of sufficient quality to contribute useful information and less overlap would be
needed. Much of this uncertainty points to the value of redundancy of sensors and the value of complementary
observing approaches, despite their potentially high cost.
While economists have extensive experience and applications in monetizing the value of potential changes in
societal outcomes (e.g. lives saved, reduced damages, improved crop yields, etc.), it is generally much more difficult
to and thus important to build more applications (1) identifying all of the potential stakeholders and potential
outcomes and (2) validly and reliably characterizing and quantifying the information value chain and how it
changes.
**9. Conclusion**
We acknowledge, as many colleagues before us (e.g. WMO, 2011a; Wulfmeyer et al., 2015; Ohring et al., 2005;
Wielicki et al., 2013), the importance of a continuous satellite record to understand solar and planetary behavior. In
this paper we focus on the development of a relatively stable data record, making full use of available satellite data,
as opposed to calibration efforts to allow a traceable record of absolute accuracy. We examine three aspects for the
merging of satellite data: identifying and quantifying an offset between two satellite records, estimating drifts
between two satellite records, and understanding the impacts of sudden changes in the data records on both offset
and drift estimates. For studies making direct use of the satellite data, either to develop a continuous record or verify
the stability of a record, the most direct control available in an observing strategy is to control the length of overlap
in the satellite records. We identify the impact of length of time of overlap on all three of these aspects of merging
satellite data and illustrate these approaches with data from two instruments used to observe solar output.
The uncertainty due to the merging of satellite records is unavoidable, but quantification of this uncertainty is
possible. In the case of identifying or verifying the offset in two satellite records, the uncertainty is inversely
proportional to the square root of the number of months of overlap. In the case of identifying or verifying the long-
term relative stability or potential drift, the number of months of overlap is inversely proportional to the drift to the
2/3 power. Both time estimates require some understanding of the variability in the overlap differences. If no
estimate of overlap variability is available, the behavior of the first few months of overlap can be evaluated to
estimate the length of overlap needed to achieve the prescribed tolerance. The impact of abrupt disruptions in the
overlap period on offsets can require up to 50% more overlap to be able to identify the offset and drift with the same
level of tolerance.
These algorithms are appropriate for a direct evaluation using only the satellite data. In some cases, particularly with
Earth observations, added benefits and challenges may exist. For instance, with Earth observations, additional in situ
and ground-based observations may be available to reduce uncertainty in satellite overlap. However, a challenge to
Earth observations is the need for direct temporal and spatial overlap, which can be difficult or impossible as
satellite observation approaches are considered within challenging budgetary constraints. All of the techniques
outlined here can be applied to identifying the level of overlap needed with existing satellite observations and next
generation observations or reference calibration satellites, such as proposed by CLARREO. In all cases, the level of
uncertainty in offsets and drifts will be determined not by the length of overlap, but in the quality of the match-up
between the reference and operational satellites. Under various constraints, choices of overlap can be optimized to
help assure climate records that are appropriate for advancing our understanding of the Earth system. The goal of
achieving the most stable observational data from existing and future observations is fundamental to understanding
the Earth and potential long-term changes. The value of this paper is the ability to estimate, either prior to satellite
launch or soon after satellite launch, the amount of time needed to achieve or verify tolerance for a stable merged
satellite record using objective criteria.
**Data and code availability:** Example data and code used will be available from the first author on request
(Betsy.Weatherhead@Colorado.edu).
**Appendix A: Comments on the usefulness of monthly data.**
The use of monthly averaged data has been common in climate studies for many years, despite obvious deficiencies
in this somewhat arbitrary choice. One deficiency is in weighting a daily value in February more highly than a daily
value from any other month simply because February has fewer number of days than, for instance, May. A second
deficiency is the lack of match-up from the monthly timeframe to the natural world: the summer solstice is not in the
center of June, but off-center meaning that the June average would contain more information on pre-solstice
conditions than post. Even non-scientific users of climate data are used to using reports such as "Climatic Normals,
World Weather Records, and Monthly Climatic Data for the World" for useful information. WMO's Guide to
Climatological Practices even suggests, "Caution is needed when data are in sub-monthly resolution..." and makes
considerable effort to coordinate climate data in a standardized manner (WMO/TD 341, 1989).
These issues are admittedly not likely of great importance, but for the reader who appreciates a great deal of caution
and respect with the use of data we offer this simple examination of the impact of monthly averages in the context of
the algorithms presented in this paper. For the example dataset used in this work, a more natural timescale is the
Carrington rotation rate of 27.2753 days. For the small study presented in this appendix, we examine how the results
of this paper would have differed if we had used the data averaged in 27-day periods as opposed to using the data
averaged by month.
The change in values from monthly values to solar rotation values, the mean, standard deviation, standard error on
the mean and autocorrelation change very little.

| | Mean (watts $m^{-2}$ $nm^{-1}$) | Standard Deviation (watts $m^{-2}$ $nm^{-1}$) | Auto-correlation | Standard Deviation of de-trended Data (watts $m^{-2}$ $nm^{-1}$) | Auto-correlation of de-trended Data | Time periods (months or solar rotation cycles) to identify an offset of 0.0008 watts $m^{-2}$ $nm^{-1}$ | Years to identify a drift of 0.00008 watts $m^{-2}$ $nm^{-1}$ $year^{-1}$ |
|---|---|---|---|---|---|---|---|
| **SIM** | | | | | | | |
| Monthly | 0.07839 | $4.78*10^{-4}$ | 0.939 | $1.528*10^{-4}$ | 0.429 | 43.6 | 3.27 |
| Solar | 0.07837 | $4.70*10^{-4}$ | 0.945 | $1.499*10^{-4}$ | 0.460 | 47.0 | 3.31 |
| **SOLSTICE** | | | | | | | |
| Monthly | 0.07907 | $3.55*10^{-4}$ | 0.890 | $1.733*10^{-4}$ | 0.544 | 13.0 | 3.94 |
| Solar | 0.07905 | $3.41*10^{-4}$ | 0.898 | $1.639*10^{-4}$ | 0.533 | 12.9 | 3.75 |
| **SOLSTICE-SIM** | | | | | | | |
| Monthly | $6.802*10^{-4}$ | $1.67*10^{-4}$ | 0.890 | $8.586*10^{-5}$ | 0.570 | 4.94 | 2.52 |
| Solar | $6.831*10^{-4}$ | $1.69*10^{-4}$ | 0.901 | $8.363*10^{-5}$ | 0.609 | 5.70 | 2.58 |

**Table A1. Fundamental descriptive values and calculations of overlap periods were calculated using monthly averaged**
**data and data that were averaged on a 27-day time period, which is a more natural timeframe for these calculations. We**
**note that little impact is observed from this small change in averaging period.**
For calculations of detecting an offset, the magnitude of variability and autocorrelation of the data were used; for the
calculations of detecting a drift, the magnitude of variability and autocorrelation of the detrended were used. The
calculations for small number of months in the final column are adjusted from a 1.96 factor to a 2.8 to account for
the uncertainty in small sample size. We note that the differences observed are remarkably small. Differences likely
would have been larger if we had used data of a shorter duration (e.g. two years of data instead of just over three
years). One note is that when the solar rotation period is used for averaging, 42 data points are derived, as opposed
to the 39 data points derived from monthly averages. This "larger number" of data points is accompanied by slightly
lower standard deviation and nearly constant auto-correlation and directly feeds into the standard error calculation.
**Appendix B: Comments on the applicability of estimation of number of years of overlap.**
For many statistical analyses commonly carried out in climate research, data are assumed to be near-Gaussian and
independent (each value is independent of the others). For environmental data, monthly averaged data are often
assumed to be auto-correlated with lag 1 month in such a manner that an AR(1) model can adequately describe the
behavior of the data once seasonal aspects are removed. As a reminder, the number of years needed to detect an
offset is estimated as:
$Months\ to\ Estimate\ an\ Offset \cong 1.96^2\ \sigma^2 \big/ _{Offset\ Limit^2} \frac{1+\varphi}{1-\varphi}$         (2)
And the number of years of overlap to detect a drift is estimated as:
$Months\ to\ Estimate\ a\ Drift \cong 12 * \left[ 1.96\ \frac{\sigma}{|trend|} * \sqrt{\frac{1+\varphi}{1-\varphi}} \right]^{2/3}$         (3)
with σ and φ as the monthly standard deviation and autocorrelation as described in the body of the paper. Because
the estimate of the number of years is dependent on these assumptions, we explicitly test the data used as an
example in this paper for illustrative purposes.
The autocorrelation, φ, is the most difficult parameter to estimate accurately in a time series, particularly when φ is
large. In the case of large autocorrelation, the time series can differ from the long-term mean for many months; if the
estimate of phi is made from a small number of points, the sample estimate of phi can be off, but the standard
deviation and mean can also be far from representative:

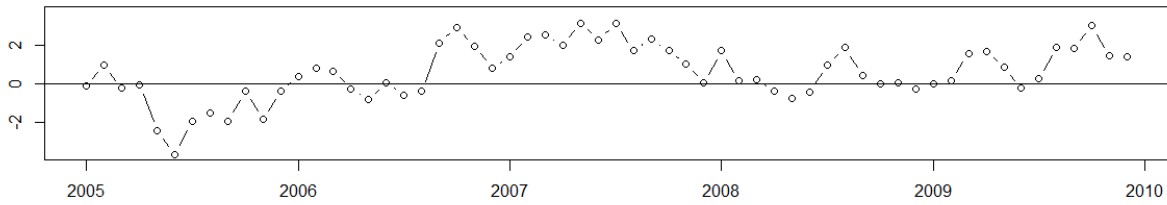

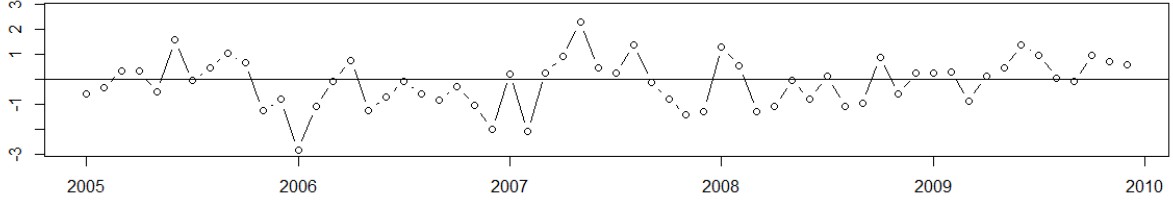

**Figure B1. Two simulated time series are shown with high autocorrelation (top plot) and low autocorrelation (bottom**
**plot).**
In situations of high autocorrelation (0.7 in the top plot of Figure B1), a time series can deviate from the long-term
mean (0 in both of these simulated time series) for many months. If a short time period is used to estimate phi, likely
phi will be under-estimated and the error on the sample mean may be farther from the true population mean
compared to a situation with low autocorrelation (0.2 in the bottom plot of Figure B1). For the data used in this
paper, the autocorrelation is estimated at 0.1 once drifts are accounted for and therefore the overlap period of six
years is more than adequate to derive a good estimate for the long-term value of phi. To test for AR(1) behavior in
the SOLSTICE-SIM monthly overlap data, we calculate a partial autocorrelation function out to fifteen terms.

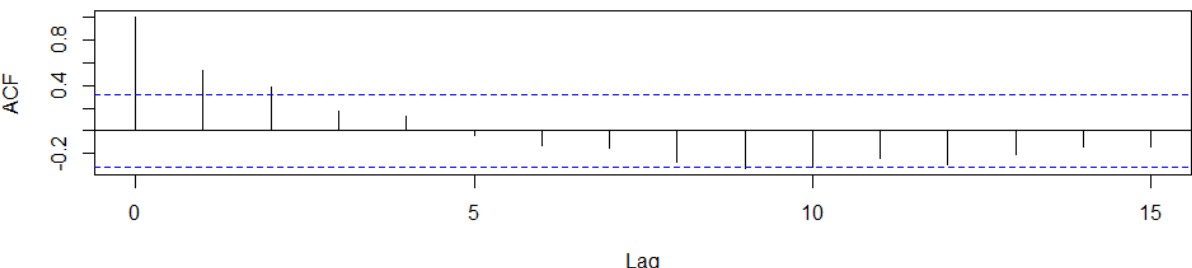

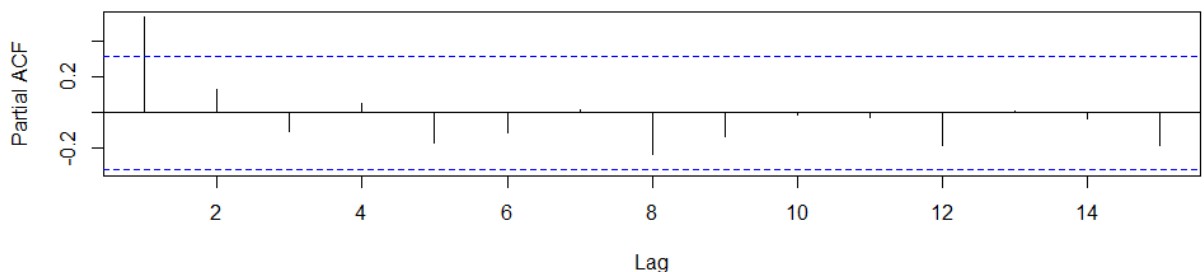

**Figure B2. The Autocorrelation Function results and Partial Autocorrelation Function results support the temporal**
**behavior of an AR(1) statistical model and therefore support the use of both Eq. (2) and Eq. (3) in the body of the paper.**
**The dashed blue lines indicate thresholds of statistical significance for results. Each lag, for this dataset represents**
**calculations based on differences of one month.**
Figure B2 shows both an AutoCorrelation Function (ACF) for the SOLSTICE-SIM data (top plot) and a Partial
Autocorrelation Function (PACF) for the SOLSTICE-SIM data (bottom plot). The plots allow confirmation that the
residuals of the data behave as an AR(1) process: the ACF shows significant autocorrelation for two months lag; the
PACF shows that there is no significant correlation once a lag 1 correlation is accounted for. The standard deviation,
$\sigma$, is assumed to represent the spread of Gaussian or normal distribution. The standard deviation calculation can be
carried out on any distribution and can be both informative and useful for many distributions. For the "number of
years" estimate to be appropriate, the assumption is that the standard deviation represents the spread in a Gaussian
distribution. For the AR(1) case, the test for Gaussian behavior is performed on the underlying interventions in the
AR(1) process, which is similar to, but not identical to the residuals observed in the overlapped data. To test for
Gaussian behavior, we compare our data to a standard Gaussian distribution in a Q-Q plot (e.g. Hamilton, 1994; Box
et al., 2015):

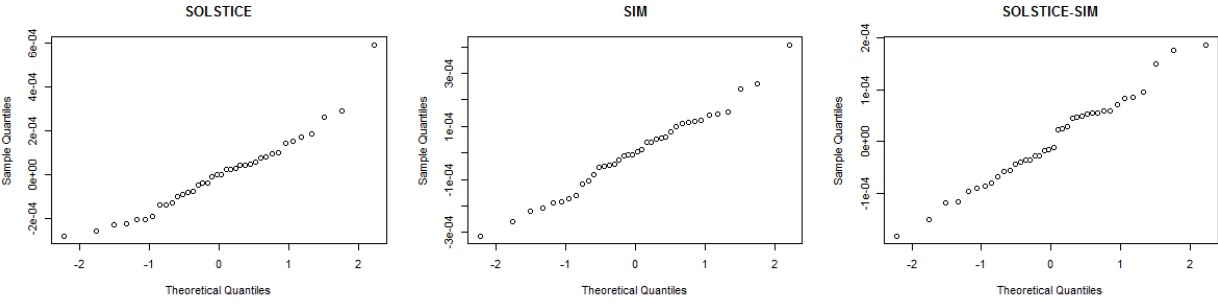

**Figure B3. These three Q-Q plots compare the monthly averaged SOLSTICE, SIM and SOLSTICE-SIM differences to**
**theoretical Gaussian distributions.**
Q-Q plots, such as those shown in Figure B3, compare the distribution from a pure Gaussian distribution to the
distribution of deseasonalized data used in this study. The roughly linear relationship demonstrated in the Q-Q plots
of B3 shows that the three datasets do behave in a close to Gaussian nature and thus the use of Eq. (2) and Eq. (3)
are supported for the analyses presented in this paper.
**Appendix C: Comments on the interpretation of time estimates.**
Any estimate of how long it will take to correctly identify a drift must be taken with some level of understanding of
how this estimate is made and what can be expected from using these estimates. Figure 5 offers estimates for a range
of times needed to estimate specific drifts, assuming no jumps occur in the record. As a reminder, this plot was
created assuming the type of overlap seen in the SOLSTICE-SIM overlap period; specifically, the calculations
assume the amount of variability and autocorrelation observed in the differences (shown in the second plot of Fig.
4). However, different observing systems are likely to have different levels of agreement.
When deriving drifts on existing data, only a single level of certainty is required: for example, what does it mean to
detect a drift? Often the community has focused on detection at a 95% confidence level or a 99% confidence level.
However, when estimating how long it will take to detect a drift, two statistical levels are required: one that
identifies what is meant by "detecting a drift" and one that identifies the likelihood that a drift will be detected in the
specified period of time, if that level of drift is the true, long-term drift in the overlap. For the first, we consider
detecting a drift to mean identifying a drift that, with 95% likelihood, is not zero, although other levels may be
considered. For the second, we consider the likelihood of detecting the drift (at the 95% confidence level) to be
50%. To be clear, we may detect the drift, if it is real, a few months earlier or a few months later than the estimated
time.
There are no error bars in Fig. 5. We'd like to begin the discussion of appropriate error bars in this section. As stated
in the previous paragraph, the data in Fig. 5 represent estimates of how long it will take to detect a specific level of
drift. If we focus on a single point, for instance the two year point that indicates a drift of $1.2*10^{-4}$ watts m$^{-2}$ nm$^{-1}$
year$^{-1}$ could be detected, it is possible that a slightly smaller drift could be detected in that two year of overlap, if the
variability happens to result in a signal-to-noise for the overlap period that is slightly more favorable. Similarly, if
the actual, underlying drift is actually five times as large ($6*10^{-4}$ watts m$^{-2}$ nm$^{-1}$ year$^{-1}$) it is highly likely the drift
would be detectable within the two years. So the "error bars" on this one point would be slightly below the current
point and would extend infinitely upward, indicating that much larger drifts could be detected in the two year period.
Extending our discussion of error bars in Fig. 5, we can similarly think in terms of horizontal error bars. Again,
focusing on the one point in Fig. 3 indicating that a drift of $1.2*10^{-4}$ watts m$^{-2}$ nm$^{-1}$ year$^{-1}$ could be detected in two
years, this drift, if it is the true underlying drift, may be detectable a few months shy of two years or may take a few
months more than two years. As stated above, the two years is a 50% likelihood of detection. It is highly likely that
such a drift could not be detected in a few months of monitoring, but it would very likely be detected in ten years of
monitoring. So, again, we have error bars that are non-standard in that they extend to the left in the plot and continue
indefinitely to the right.

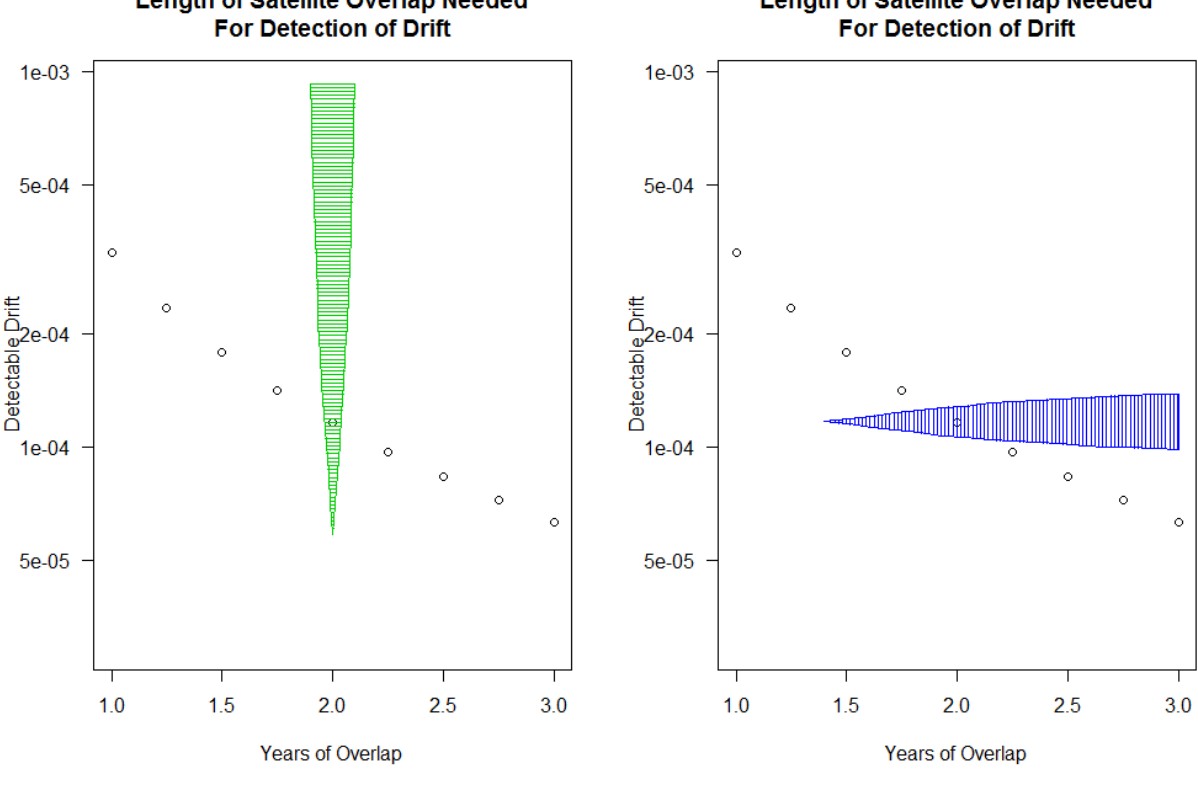

**Figure C1. Estimates of how long it will take to detect a drift can be interpreted as the likely time needed. Depending on**
**variability present, even small drifts can be detected (although with less than 50% likelihood of detection), probability**
**indicated by the width of the green area. For a given drift level, there is a chance that the drift can be detected in less than**
**the number of years indicated, although that likelihood is less than 50% for time less than the times indicated, and the**
**probability indicated by the blue area.**
If we want to express this uncertainty of likelihood of detection in a visual manner, we could employ two
dimensional error bars, similar to violin plots which are often employed to express variable information. Figure C1
shows the likelihood of detecting a particular drift with two years of overlap. For drifts considerably smaller than
$1.2*10^{-4}$ watts m$^{-2}$ nm$^{-1}$ year$^{-1}$, the likelihood of detection with a two year overlap is represented by the width of the
green area. Larger drifts can be detected with higher likelihood. Figure C1 shows the likelihood of a true drift of
$1.2*10^{-4}$ watts m$^{-2}$ nm$^{-1}$ year$^{-1}$ being detected in less than two years. The height of the blue bar indicates the
likelihood, with the linear scale being defined such that the likelihood of detection being 50% at two years;
considerably higher likelihood of detection is indicated with more years of overlap. There is also a small, but less
than 50% likelihood that the true drift might be detected in less than two years; again, the height of the blue bars
indicates the likelihood of detection.
**Appendix D: Mathematical structure for optimizing overlap decision choices.**
Treating the value of information derived from satellite data as a public good (e.g. weather forecasts and climate
services have non-rival and non-excludable characteristics which define public goods in economic theory), total
societal benefits is the sum of the benefits realized by all users of the information. Net benefits (NB) is the
difference between total benefits (TB) and total costs (TC).
$$\mathbf{NB}=\mathbf{TB}\{IVC[q(o)]\}\text{-}\mathbf{TC}[o] \qquad\qquad (D1)$$
For purposes of the current discussion we take total costs to simply be a function of the temporal overlap in satellite
observations ($o$). Total costs are an increasing function of $o$ (i.e., the total costs increase the more the overlap
period). On the other hand, total benefits are a more complicated function of the entire process of information
creation, communication, use, and decision making (labeled IVC for the Information Value Chain). The benefits of
the IVC process are considered to be a function of the quality of the information, $q$, which itself is a function of the
temporal overlap in satellite observations ($o$). This is a highly non-linear process and not even necessarily increasing
in $q(o)$ as may commonly be assumed.
If the objective of "optimizing" satellite observation overlaps is to achieve a specific quality standard, economists
would approach this as a cost-effectiveness issue. In this case, the objective is to minimize the costs to achieve the
exogenously determined standard. There is no consideration of benefits in this case, $o$ is set to achieve $\overline{q}$ so
$\mathbf{TB}\{IVC[\overline{q}(o)]\}$. The outcome is not necessarily societally optimal – the standard could be too strict in which
case net benefits could be negative or the standard could be too lax in which case it would be possible to improve
societal outcomes by increasing the observational overlap period and increase net benefits.[3]
If instead of a fixed standard $\overline{q}$, the objective is to maximize net societal benefit then $o$ is chosen to maximize NB.
Maximal societal benefits can be identified as a function of the temporal overlap by taking the first derivative of the
NB formula with respect to $o$.[4]
$$\partial\mathbf{NB}/\partial o =\partial\,\mathbf{TB}\{IVC[q(o)]\}/\partial o\,\text{-}\partial\mathbf{TC}/\partial o = \frac{\partial\mathbf{TB}}{\partial IVC}\frac{\partial\mathbf{IVC}}{\partial q}\frac{\partial q}{\partial o}\text{-}\partial\mathbf{TC}/\partial o = \mathbf{MB}_o - \mathbf{MC}_o \qquad (D2)$$
As the terms on the right indicate, the maximum societal benefits are achieved when the marginal benefits ($MB_o$) of
a change in the overlap period are equal to the marginal costs ($MC_o$) of the overlap.[5] The marginal costs of the
overlap period, $\partial\mathbf{TC}/\partial o$, are likely to include additional costs of data collection, assimilation, storage, and
analysis and may be a fairly linear function of the length of the overlap period. The marginal benefits of increasing
the overlap period though are represented as a more complex relationship between the overlap period and quality of
information ($\partial\mathbf{q}/\partial o$), how changes in information quality manifest through in the information value chain (
$\partial\mathbf{IVC}/\partial q$), and how changes in the quality of information provided to decision makers may manifest themselves
in potential outcomes ($\partial\mathbf{TB}/\partial IVC$). While economists have extensive experience and applications in monetizing
the value of potential changes in societal outcomes (e.g. lives saved, reduced damages, improved crop yields in
agricultures, etc.), it is generally much more difficult to (1) identify all of the potential stakeholders and potential
outcomes and (2) validly and reliably characterize and quantify the information value chain or how it changes.

---

[3] Rather than taking the standard as a given, the question could be how to set the standard to achieve a societally
"optimal" outcome in terms of maximizing societal benefits. This is essentially equivalent to the unconstrained
optimization of NB.

[4] For now we don't discuss second order conditions for maximization. See Morss et al. (2005) for further
clarification on second order conditions for net benefit maximization which relate specifically to the shapes of the
cost and benefits functions.

[5] In economics notation it is common to use the subscript to indicate the relevant factor under discussion – in this
case $MB_o$ refers to the marginal benefits of $o$, the observational overlap period.

To dependably quantify societal benefits from satellite observations requires understanding the complex relationship
between individual instruments, data streams, modeling, communication, decision making, and potential and actual
societal outcomes. This requires understanding stakeholders and processes of information creation, transformation,
transmission, and use in decision making along the entire information value chain through multiple stakeholders
with a variety of objectives, resources, and constraints. Cooke et al. (2014) develop an illustrative example of such
an analysis using the social costs of carbon (SCC) as a measure of societal benefits and a hypothetical decision
framework (i.e., a tipping point that would lead to global climate impact mitigation efforts).

1  **Team list**: E. Weatherhead, J. Harder, E. Araujo-Pradere, G. Bodeker, J. M. English, L. E. Flynn, S. M. Frith, J. K.
2  Lazo, P. Pilewskie, T. Woods, M. Weber,

4  **Author contribution:** Jerald Harder conceived of the idea for this paper and the general techniques to be used; he
5  also provided the data for analysis. Elizabeth Weatherhead carried out the calculations, including adapting statistical
6  approaches to the problem and oversaw the writing. Eduardo Araujo worked on the calculations and identified
7  stability requirements for solar and environmental data. Larry Flynn supplied input on GSICS and complementary
8  methods of verifying stability in satellite records as well as cross verification of results, including those presented in
9  Table 1. Stacey Frith and Mark Weber provided valuable input on merging of satellite records and applicability of
10  results to developing long-term records. Greg Bodeker offered data and insights into merging of satellite
11  temperature datasets. Jeff Lazo, in consultation with the other co-authors, provided the optimization methodology
12  and economic analysis. Thomas Woods and Peter Pilewskie, as respective Principal Investigators for the NASA
13  SORCE and the upcoming TSIS missions advised on the requirements and needs for mission overlap for the solar
14  spectral irradiance climate record. Jason English supplied input on applicability to climate studies.

15  **Competing interests:** There are no known competing interests for this work. The authors declare that they have no
16  conflict of interest.

17  **Disclaimer:** None

18  **Acknowledgements:** We would like to acknowledge the support of NASA for the production of the data used in
19  this project. NASA contract NAS5-97045, and Miami-Dade College for their research support. M. Weber
20  acknowledges the support of the DFG Research Unit SHARP (Stratospheric Change and its Role for Climate
21  Prediction).

22  **References**

23  Adams, C., Bourassa, A. E., Sofieva, V., Froidevaux, L., McLinden, C. A., Hubert, D., Lambert, J.-C., Sioris, C. E.,
24    and Degenstein, D. A.: Assessment of Odin-OSIRIS ozone measurements from 2001 to the present using
25    MLS, GOMOS, and ozonesondes, Atmos. Meas. Tech., 7, 1, 49-64, 2014.

26  Araujo-Pradere, E. A., Buresova, D., and Fuller-Rowell, T. J.: Initial results of the evaluation of IRI hmF2
27    performance for minima 22-23 and 23-24, Adv. Space Res., doi: 10.1016/j.asr.2012.02.010, 2012.

28  Araujo-Pradere, E. A., Fuller-Rowell, T. J., and Bilitza, D.: Ionospheric variability for quiet and perturbed
29    conditions, Adv. Space Res., 34(9), 1914-1921, doi: 10.1016/j.asr.2004.06.007, 2004.

30  Araujo-Pradere, E. A., Redmon, R., Fedrizzi, M., Viereck, R., and Fuller-Rowell, T. J.: Some characteristics of the
31    ionospheric behavior during solar cycle 23/24 minimum, Solar Phys., The Sun–Earth Connection near
32    Solar Minimum, edited by M. M. Bisi, B. Emery, and B. J. Thompson, doi: 10.1007/s11207-011-9728-3,
33    Springer, 2011.

34  ASIC3: Achieving Satellite Instrument Calibration for Climate Change (ASIC3). Ohring, G. (Ed.), Available at:
35    http://www.star.nesdis.noaa.gov/star/documents/ASIC3-071218-webversfinal.pdf, 2007.

36  BenMoussa, A., and forty additional authors: On-orbit degradation of solar instruments, Solar Phys., 288, 389-434,
37    doi: 10.1007/s11207-013-0290-z, 2013.

38  Best, F., Adler, A. D. P., Ellington, S. D., Thielman, D. J., and Revercomb, H. E.: On-orbit absolute calibration of
39    temperature with application to the CLARREO mission, Proc. SPIE, doi: 10.1117/12.795457, 2008.

40  Bodeker, G. E., Scott, J. C., Kreher, K., and McKenzie, R. L.: Global ozone trends in potential vorticity coordinates
41    using TOMS and GOME intercompared against the Dobson network: 1978–1998, J. Geophys. Res. Atmos.,
42    106(D19), 23029-23042, 2001.

43  Bourassa, A. E., Degenstein, D. A., Randel, W. J., Zawodny, J. M., Kyrölä, E., McLinden, C. A., Sioris, C. E., and
44    Roth, C. Z.: Trends in stratospheric ozone derived from merged SAGE II and Odin-OSIRIS satellite
45    observations, Atmos. Chem. Phys., 14, 6983-6994, doi: 10.5194/acp-14-6983, 2014.

Box, G. E. P., Jenkins, G. M., Reinsel, G. C., and Ljung, G. M.: Time series analysis: forecasting and control, 4th edition, John Wiley & Sons, 2015.

Brown, S.: Maintaining the long-term calibration of the Jason-2/OSTM advanced microwave radiometer through intersatellite calibration, IEEE T. Geosci. and Remote Sensing, 51, 1531-1543, 2013.

Chander, G., Helder, D. L., Aaron, D., Mishra, N., and Shrestha, A. K.: Assessment of spectral, misregistration and spatial uncertainties inherent in the cross-calibration study, IEEE T. Geosci. and Remote Sensing, 51, 1282-1296, 2013a.

Chander, G., Hewison, T. J., Fox, N., Wu, X., Xiong, X., and Blackwell, W. J.: Overview of intercalibration of satellite instruments, IEEE T. Geosci. and Remote Sensing, 51, 1056-1080, 2013b.

Christy, J. R., Spencer, R. W., and Braswell, W. D.: MSU tropospheric temperatures: Dataset construction and radiosonde comparisons, J. Atmos. and Ocean. Tech., 17, 9, 1153-1170, 2000.

Christy, J. R., Spencer, R. W., and Lobl, E. S.: Analysis of the merging procedure for the MSU daily temperature time series, J. Climate, 11, 8, 2016-2041, 1998.

Christy, J. R.: Temperature above the surface layer in long-term climate monitoring by the global climate observing system edited by T. Karl, Vol. 31, Issue 2, pp 455-474, Springer, Netherlands, 1995.

Cooke, R., Wielicki, B. A., Young, D. F., and Mlynczak, M. G.: Value of information for climate observing systems, Environ. Syst. Decis., 34, 98, doi: 10.1007/s10669-013-9451-8, 2014.

Donaldson, D., and Storeygard, A.: The view from above: applications of satellite data in economics, J. Econ. Perspect, 30(4), 171-98, doi: 10.1257/jep.30.4.171, 2016.

Ducre'-Robitaille, J.-F., Vincent, L. A., and Boulet, G.: Comparison of techniques for detection of discontinuities in temperature series, Int. J. of Climatol., doi: 10.1002/joc.924, 2003.

Dudok de Wit, T., Kretzschmar, M., Aboudarham, J., Amblard, P.-O., Auchère, F., and Lilensten, J.: Which solar EUV indices are best for reconstructing the solar EUV irradiance?, Adv. Space Res., 42, 5, 903-911, 2008.

Dudok de Wit, T.: A method for filling gaps in solar irradiance and solar proxy data, Astron. Astrophys, 533, A29, 2011.

Eckert, E., T. von Clarmann, M. Kiefer, G. P. Stiller, S. Lossow, N. Glatthor, D. A. Degenstein, L. Froidevaux, S. Godin-Beekmann, T. Leblanc, S. McDermid, M. Pastel, W. Steinbrecht, D. P. J. Swart, K. A. Walker, and P. F. Bernath. Drift-corrected trends and periodic variations in MIPAS IMK/IAA ozone measurements, Atmos. Chem. Phys., 14, 2571-2589, 2014.

Feldman, D. R., Algieri, C. A., Ong, J. R., and Collins, W. D.: CLARREO shortwave observing system simulation experiments of the twenty-first century: simulator design and implementation, J. Geophys. Res., 116 (D10), doi: 10.1029/2010JD015350, 2011.

Feldman, D. R., Collins, W. D., and Paige, J. L.: Pan-spectral observing system simulation experiments of shortwave reflectance and longwave radiance for climate model evaluation, Geoscientific Model Development, 8, 1943-1954, doi: 10.5194/gmd-8-1943, 2015.

Fioletov, V. E., G. E. Bodeker, A. J. Miller, R. D. McPeters, and R. Stolarski, 2002: Global and zonal total ozone variations estimated from ground-based and satellite measurements: 1964–2000. J. Geophys. Res., 107, 4647, doi:10.1029/2001JD001350.

Fox, N., Green, P., Brindley, H., Russell, J., Smith, D., Lobb, D., Cutter, M., and Barnes, A.: TRUTHS (Traceable Radiometry Underpinning Terrestrial and Helio Studies): A mission to achieve climate quality data, Proc. 'ESA Living Planet Symposium 2013', Edinburgh, ESA SP-722, December 2013.

Free, M., Durre, I., Aguilar, E., Seidel, D., Peterson, T. C., Eskridge, R. E., Luers, J. K., Parker, D., Gordon, M., Lanzante, J., Klein, S., Christy, J., Schroeder, S., Soden, B., McMillin, L. M., and Weatherhead, E.: Creating Climate Reference Datasets: CARDS Workshop on adjusting radiosonde temperature data for climate monitoring, Bull. Am. Meteorol. Soc., doi: http://dx.doi.org/10.1175/1520-0477(2002)083<0891:CCRDCW>2.3.CO;2, 2002.

Frith, S. M., Kramarova, N. A., Stolarski, R. S., McPeters, R. D., Bhartia, P. K., and Labow, G. J.: Recent changes in total column ozone based on the SBUV Version 8.6 merged ozone data set, J. Geophys. Res. Atmos., 119(16), 9735-9751, 2014.

Fruit, M., Gusarov, A., and Doyle, D.: Testing and qualification of optical glasses for use in a space radiation environment; the advantages and pitfalls of using a parametric approach, Proc. SPIE Int. Soc. Opt. Eng., 4823, 132, 2002.

Fröhlich C.: Evidence of a long-term trend in total solar irradiance, Astron. Astrophys., 501, L27-L30, doi: 10.1051/0004-6361/200912318, 2009.

Hamilton, J. D.: Time series analysis, vol. 2. Princeton: Princeton University Press, 1994.

Harder J. W., Lawrence, G., Fontenla, J., Rottman, G. J., and Woods, T. N.: The Spectral Irradiance Monitor: scientific requirements, instrument design, and operation modes, Solar Phys., 230, 141-167, 2005a.

Harder J. W., Fontenla, J., Lawrence, G., Woods, T., and Rottman, G.: The Spectral Irradiance Monitor: Measurement (SIMM) equations and calibration, Solar Phys., 230, 169-204, 2005b.

Harder, J. W., Thuillier, G., Richard, E. C., Brown, S. W., Lykke, K. R., Snow, M., McClintock, W. E., Fontenla, J. M., Woods, T. N., and Pilewskie, P.: The SORCE SIM Solar Spectrum: comparison with recent observations, Solar Phys., 263, 3-24, doi: 10.1007s11207-010-9555-y, 2010.

Hewison, T. J., Wu, X., Yu, F., Tahara, Y., Hu, X., Kim, D., and Koenig, M.: GSICS Inter-calibration of infrared channels of geostationary imagers using Metop/IASI, IEEE T. Geosci. and Remote Sensing, 15 (3), 2013.

Hood, L. L., and Zhou, S.: Stratospheric effects of 27-day solar ultraviolet variations: An analysis of UARS MLS ozone and temperature data, J. Geophys. Res., 103(D3), 3629-3638, doi: 10.1029/97JD02849, 1998.

Hubert, D., and 39 co-authors, Ground-based assessment of the bias and long-term stability of 14 limb and occultation ozone profile data records, AMT, 2016,

Hurrell, J., and Trenberth, K.: Spurious trends in satellite MSU temperatures from merging different satellite records, Nature, 386, doi: 10.1038/386164a0, 1997.

Jaxk, R., Chen, J. L., Wang, X. L., Lund, R., and Lu, Q. Q.: A review and comparison of changepoint detection techniques for climate data, J. Appl. Meteorol. Climat., doi: http://dx.doi.org/10.1175/JAM2493.1, 2007.

JCGM – Joint Committee for the Guides in Metrology, JCGM 101:, available at: http://www.bipm.org/en/publications/guides/gum.html, 2008.

Karl, T. R., and Williams, C. N.: An approach to adjusting climatological time series for discontinuous inhomogeneities, J. Appl. Meteorol. Climat., doi: http://dx.doi.org/10.1175/1520-0450(1987)026<1744:AATACT>2.0.CO;2, 1987.

Karl, T. R., Williams Jr., C. N., Young, P. J., and Wendland, W. M.: A model to estimate the time of observation bias associated with monthly mean maximum, minimum, and mean temperature for the United States, J. Climate Appl. Meteorol., 25, 145-160, 1986.

Laxminarayan, R., and Macauley, M. K. (Eds.): The value of information: methodological frontiers and new applications in environment and health. Springer Netherlands. 304 pages, doi: 10.1007/978-94-007-4839-2, 2012.

Loeb, N. G., Nanalo-Smith, N., Su, W., Shankar, M., and Thomas, S.: CERES top-of-Atmosphere Earth radiation budget climate data record: Accounting for in-orbit changes in instrument calibration, Remote Sens., 8, 182, doi: 10.3390/rs8030182, 2016.

Logan, J. A., Staehelin, J., Megretskaia, I. A., Cammas, J. P., Thouret, V., Claude, H., & Fröhlich, M. Changes in ozone over Europe: Analysis of ozone measurements from sondes, regular aircraft (MOZAIC) and alpine surface sites. Journal of Geophysical Research: Atmospheres, 117(D9), 2012

Lukashin, C., Wielicki, B. A., Young, D. F., Thome, K., Jin, Z., and Sun, W.: Uncertainty estimates for imager reference inter-calibration with CLARREO reflected solar spectrometer, IEEE T. Geosci. and Remote Sensing, 2013.

Macauley, M. K.: The value of information: measuring the contribution ofspace-derived earth science data to resource management, Space Policy, 22(4), 274-282, 2006.

MacDonald, A. E.: A global profiling system for improved weather and climate prediction, Bull. Am. Met. Soc., 86, 1747, doi: 10.1175/BAMS-86-12-1747, 2005.

Marchenko, S. V., and DeLand, M. T.: Solar spectral irradiance changes during cycle 24, The Astrophys. J., 789, 2, 117, doi: 10.1088/0004-637X/789/2/117, 2014.

McClintock W. E., Rottman G. J., and Woods T. N.: Solar-Stellar Irradiance Comparison Experiment II (SOLSTICE II): Instrument concept and design, Solar Phys., 230, 225-258, 2005a.

McClintock W. E., Snow, M., and Woods, T. N.: Solar-Stellar Irradiance Comparison Experiment II (SOLSTICE II): Pre-launch and on-orbit calibrations, Solar Phys., 230, 259-294, 2005b.

Mitchell, T. D., and Jones, P. D.: An improved method of constructing a database of monthly climate observations and associated high-resolution grids, Int. J. of Climatol., doi 10.1002/joc.1181, 2005.

Morss, R., Miller, K. A., and Vasil, M. S.: A systematic economic approach to evaluating public investment in observations for weather forecasting, Mon. Weather Rev., 133, 374-388, 2005.

National Research Council, Issues in the integration of research and operational satellite Ssystems for climate research: Part I. science and design. Committee on Earth Studies, Space Studies Board, NRC, 2000a.

National Research Council, Reconciling observations of global temperature change, National Academy Press, Washington, D.C., 2000b.

NISTIR 7047, Satellite Instrument Calibration for Measuring Global Climate Change. Ohring, G., Wielicki, B., Spencer, R., Emery, W., Datla, R. (Eds). Available at: https://www.nist.gov/sites/default/files/documents/pml/div685/pub/nistir7047.pdf, 2004.

Ohring, G., Wielicki, B., Spencer, R., Emery, B., and Datla, R.: Satellite instrument calibration for measuring global climate change: report of a workshop, Bull. Am. Meteorol. Soc., 86, 1303-13, doi: 10.1175/BAMS-86-9-1303, 2005.

Pagaran J., Harder, J. W., Webber, M., Floyd, L. E., and Burrows, J. P.: Intercomparison of SCIAMACHY and SIM vis-IR irradiance over several solar rotational timescales, Astron. Astrophys., 528, A67, 2011.

Penckwitt, A. A., G. E. Bodeker, P. Stoll, J. Lewis, T. von Clarmann, and A. Jones. Validation of merged MSU4 and AMSU9 temperature climate records with a new 2002–2012 vertically resolved temperature record, submitted to AMTD 8, 235–267, 2015.

Peterson, T. C., Easterling, D. R., Karl, T. R., Groisman, P., Nicholls, N., Plummer, N., Torok, S., Auer, I., Boehm, R., Gullett, D., Vincent, L., Heino, R., Tuomenvirta, H., Mestre, O., Szentimrey, T., Salinger, J., Forland, E. J., Hanssen Bauer, I., Alexandersson, H., Jones, P., and Parker, D.: Homogeneity adjustments of in situ atmospheric climate data: a review, Int. J. Climatol., 18, 1493-1517, 1998.

Rahpoe, N., Weber, M., Rozanov, A. V., Weigel, K., Bovensmann, H., Burrows, J. P., and Laeng, A.: Relative drifts and biases between six ozone limb satellite measurements from the last decade, Atmos. Meas. Tech., 8, 10, 4369-4381, 2015.

Randel, W. J., and Thompson, A. M.: Interannual variability and trends in tropical ozone derived from SAGE II satellite data and SHADOZ ozonesondes, J. Geophys. Res., 116, D07303, doi: 10.1029/2010JD015195, 2011.

Salby, M., and Callaghan, P.: Sampling error in climate properties derived from satellite measurements: consequences of undersampled diurnal variability, J. Climate, 10, 18-36, 1997.

Santer, B. D., Wigley, T. M. L., Meehl, G. A., Wehner, M. F., Mears, C., Schabel, M., Wentz, F. J., Ammann, C., Arblaster, J., Bettge, T., Washington, W. M., Taylor, K. E., Boyle, J. S., Brüggemann, W., and Doutriaux, C.: Influence of satellite data uncertainties on the detection of externally forced climate change, Science, doi: 10.1126/science.1082393, 2003.

Schrijver, C. J., Livingston, W. C., Woods, T. N., and Mewaldt, R. A.: The minimal solar activity in 2008–2009 and its implications for long term climate modeling, Geophys. Res. Lett., 38, 6, 2011.

Slater, P. N., Biggar, S. F., Thome, K. J., Gellman, D. I., and Spyak, P. R.: Vicarious radiometric calibrations of EOS sensors, J. Atmos. and Ocean. Tech., 13, 2, 349-359, 1996.

Smith, T. M., Reynolds, R. W., Peterson, T. C., and Lawrimore, J.: Improvements to NOAA's historical merged land-ocean surface temperature analysis (1880-2006), J. Climate, 21, 10, 2283-2296, 2008.

Snow, M., McClintock, W. E., Rottman, G., and Woods, T. N.: Solar-Stellar Irradiance Comparison Experiment II (SOLSTICE II): Examination of the Solar-Stellar Comparison Technique, Solar Phys., 230, 295-324, 2005.

Staehelin, J., Kerr, J., Evans, R., and Vanicek, K.: Comparison of total ozone measurements of Dobson and Brewer spectrophotometers and recommended transfer functions, Tech. Rep., WMO, World Meteorological Organization Global Atmosphere Watch (WMO-GAW) Report 149. http://library.wmo.int/pmb_ged/wmo-td_1147.pdf, 2003.

Stolarski, R., and Frith, S.: Search for evidence of trend slow-down in the long-term TOMS/SBUV total ozone data record: the importance of instrument drift uncertainty, Atmos. Chem. Phys., 6, 4057-4065. doi: 10.5194/acp-6-4057, 2006.

Tegtmeier, S., Hegglin, M. I., Anderson, J., Bourassa, A., Brohede, S., Degenstein, D., ... & Jones, A. SPARC Data Initiative: A comparison of ozone climatologies from international satellite limb sounders. Journal of Geophysical Research: Atmospheres, 118(21), 2013.

Thorne, P. W., Parker, D. E., Christy, J. R., and Mears, C. A.: Uncertainties in climate trends: lessons from upper-air temperature records, Bull. Am. Meteorol. Soc., 86, 10, 1437-1442, 1398-1399, 2005.

Thuillier G., Floyd, L., Woods, T. N., Cebula, R., Hilsenrath, E., Hers, M., and Labs, D.: Solar irradiance reference spectra for two solar active levels, Adv. Space Res., 34, 256-261, 2004.

Tobin, D., Holz, R., Nagle, F., and Revercomb, H.: Characterization of the climate absolute radiance and refractivity observatory (CLARREO) ability to serve as an infrared satellite intercalibration reference, J. Geophys. Res., 121 (8), 2016.

Toohey, M., Hegglin, M. I., Tegtmeier, S., Anderson, J., Añel, J. A., Bourassa, A., and Brohede, S.: Characterizing sampling biases in the trace gas climatologies of the SPARC Data Initiative, J. Geophys. Res. Atmos., 118, 20, 2013.

Toohey, M. and von Clarmann, T.: Climatologies from satellite measurements: the impact of orbital sampling on the standard error of the mean. Atmo. Meas.Tech.,6(4), pp.937-948, 2013.

Viereck, R. A., Floyd, L. E., Crane, P. C., Woods, T. N., Knapp, B. G., Rottman, G., Weber, M., Puga, L. C., and DeLand, M. T.: A composite Mg II index spanning from 1978 to 2003, Space Weather, 2, S10005, doi: 10.1029/2004SW000084, 2004.

Vincent, L. A., Zhang, X., Bonsal, B. R., and Hogg, W. D.: Homogenization of daily temperatures over Canada, J. Climate, 15, 1322‑1334, 2002.

Vincent, L.: A Technique for the identification of inhomogeneities in Canadian temperature series, J. Climate, doi: http://dx.doi.org/10.1175/1520-0442(1998)011<1094:ATFTIO>2.0.CO;2, 1998.

Weatherhead, E. C., Reinsel, G. C., Tiao, G. C., Jackman, C. H., Bishop, L., Hollandsworth Frith, S. M., DeLuisi, J., Keller, T., Oltmans, S. J., Fleming, E. L., Wuebbles, D. J., Kerr, J. B., Miller, A. J., Herman, J., McPeters, R., Nagatani, R. M., and Frederick, J. E.: Detecting the recovery of total column ozone, J. Geophys. Res. Atmos., 105, D17, doi: 10.1029/2000JD900063, 2000.

Weatherhead, E. C., Reinsel, G. C., Tiao, G. C., Meng, X. L., Choi, D., Cheang, W. K., Keller, T., DeLuisi, J., Wuebbles, D. J., Kerr, J. B., Miller, A. J., Oltmans, S. J., and Frederick, J. E.: Factors affecting the detection of trends: statistical considerations and applications to environmental data, J. Geophys. Res., 103 D14, 1998.

Weatherhead, E. C., Tiao, G. C., Reinsel, G. C., Frederick, J. E., DeLuisi, J. J., Choi, D., & Tam, W. K. Analysis of
long-term behavior of ultraviolet radiation measured by Robertson-Berger meters at 14 sites in the United
States. Journal of Geophysical Research: Atmospheres, 102(D7), 8737-8754, 1997.
Weatherhead, E. C., Wielicki, B., and Ramaswamy, V.: Climate observing system simulation experiments. AGU
Fall Meeting, San Francisco, California, Dec. 14-18, 2015.
Weber, M., W. Steinbrecht, C. Roth,M. Coldewey-Egbers, D. Degenstein, V.E. Fioletov, S. M. Frith, L. Froidevaux,
J. de Laat, C. S. Long, D. Loyola, and J. D. Wild: [Global Climate] Stratospheric ozone [in "State of the
Climate in 2015"], Bull. Amer. Meteor. Soc., 97, S49–S51, 2016a.
Weber, M., S. Dikty, J. P. Burrows, H. Garny, M. Dameris, A. Kubin, J. Abalichin, and U. Langematz, The Brewer-
Dobson circulation and total ozone from seasonal to decadal time scales, Atmos. Chem. Phys., 11, 11221-
11235, doi:10.5194/acp-11-11221-2011, 2011.
Weber, M., Rahpoe, N., and Burrows, J.: Stability requirements on long-term (satellite) ozone observations and their
implications for trend detection, QOS, Edinbrugh, September, 2016b.
Wentz, F. J., and Schabel, M.: Effects of orbital decay on satellite-derived lower-tropospheric temperature trends.
Nature, 394, 661-664, 1998.
Wielicki, B. A., Young, D. F., Mlynczak, M. G., Thome, K. J., Leroy, S., Corliss, J., and Anderson, J. G.: Achieving
climate change absolute accuracy in orbit, Bull. Am. Meteorol. Soc., 94, 10, 1519-1539, 2013.
Willson, R. C., and Hudson, H. S.: The sun's luminosity over a complete solar cycle, Nature, 351, 42-44, 1991.
Willson, R. C., and Mordvinov, A. V.: Secular total solar irradiance trend during solar cycles 21–23, Geophys. Res.
Lett., 30(5), 1199, doi: 10.1029/2002GL016038, 2003.
WMO Guide to Climatological Practices WMO-No. 100, ISBN 978-92-63-10100-6. Available at
http://www.wmo.int/pages/prog/wcp/ccl/guide/documents/WMO_100_en.pdf, 2011a.
WMO Guide to Climatological Practices, Third edition, Technical specification for the evolution and future hosting
of the WMO Database of Observational user requirements and observing system capabilities, 2011b.
WMO/TD‑No. 341, Calculation of Monthly and Annual 30‑years standard normals (prepared by a meeting of
experts, Washington, March 1989) [WCDP‑No. 10], 1989.
WMO, UNEP. Scientific Assessment of Ozone Depletion: 2014.
Wu, X., Hewison, T., and Tahara, Y.: GSICS GEO-LEO intercalibration: baseline algorithm and early results, SPIE
Proc., 7456, doi: 10.1117/12.825460, 2009.
Wulfmeyer, V., Hardesty, M., Turner, D., Behrendt, A., Cadeddu, M., Di Girolamo, P., Schlüssel, P., van Baelen, J.,
and Zus, F.: A review of the remote sensing of lower-tropospheric thermodynamic profiles and its
indispensable role for the understanding and the simulation of water and energy cycles, Rev. Geophys., doi:
10.1002/2014RG000476, 2015.
Zou, C. Z., and Qian, H.: Stratospheric temperature climate data record from merged SSU and AMSU-A
Observations, J. Atmos. and Ocean. Tech., 2016.