# Peer review of "How long do satellites need to overlap? Evaluation of climate data stability from overlapping satellite records"

_Atmospheric Chemistry and Physics, 2016_

## Referee Comment (RC1) · Anonymous Referee #1 · 27 Feb 2017

General comments

This paper examines the issue of quantifying differences in satellite measurements sets based on overlapping measurement periods, addressing the question of how long overlap periods need to be to accurately estimate offsets and drifts between instruments. A few general formulas are presented to calculate required overlap periods for given desired precision requirements of offset and drift estimation, and examples are presented.

While the issues discussed are definitely relevant to the construction of long-term atmospheric data records, this paper does not actually directly deal with any atmospheric measurements, instead focusing on examples based on satellite measurements of

solar irradiance. As a result, the fit between this paper and ACP (or even AMT for that matter) is somewhat questionable. Readers of ACP would likely benefit greatly from examples using actual atmospheric data, and given the ready availability of multi-instrument data sets like stratospheric ozone (e.g., Tegtmeier et al., 2013), it wouldn't be hard to include such examples. Of course, atmospheric data, with the temporal and spatial variability that comes with it, may present some additional complications to the analysis (which is mentioned in passing in the manuscript), but a discussion of these complications seems warranted in such a paper if it truly wants to address the analysis (and merging) of atmospheric data.

The utility of this paper to atmospheric community could also be improved by a fuller description of the general implications of the analysis before descending to the focused solar irradiance example. For example, Fig 3 displays the detectable drifts in the solar irradiance data sets as a function of years of overlap, which suggests a general form of the solution, but won't provide any quantitative information to anyone working with other data sets. Instead, a plot of ratio of drift to variability as a function of n (perhaps for different sample values of autocorrelation) would be directly relevant to users of other data sets.

Stylistically, I found the paper repetitious in places, often returning to discussions of issues that aren't, in my opinion, of central importance. For example, the issue of requirements (or the desire) for self-calibrating, consistent systems for atmospheric measurements is often brought up, but this paper deals specifically with techniques to deal with situations where measurements are not self-calibrated. This point can be made succintly in the introduction, and thereafter neglected, at least until the discussion and conclusions.

Specific comments:

Pg 1, l27: "offset or a drift in the offsets": If "offset" is singular in the first case, then "drift in the offset" seems more appropriate. But the sentence was confusing to me at

first, and I wonder if just "drift" is easier to understand.

Pg 1, l37-38: "may also benefit. . ." this issue is not dealt with in any substantial way in the paper, so this statement's inclusion in the abstract seems superfluous.

Pg 2, l16: "tying data to absolute reference standards with the intent of developing traceability to reference standards" sounds a little tautological.

Pg 2, l17: It's not clear to me why reference standards are brought into the argument here, is the point that if one of the two overlapping measurement sets is a standard, then you can extend a standard through identification of an offset and drift in the second instrument?

Pg 2, l36: Do "wavelength scale corrections" etc. really help instrument scientists understand the fundamental observations? Or does an understanding of the fundamental observations allow for valid corrections?

Pg 3, l15: Does removing a bias affect the precision of the merged data set? And, does one really need to remove a bias to identify a drift? If you look at changes with time (time derivatives) the absolute value doesn't matter.

Pg 3, l28: It would seem that the paper is of interest to a wider group than just the users of merged data sets, specifically to the creators of merged data sets.

Pg 3, ll40: The first two paragraphs of Sec 1.1 have no apparent specific connection to "Offsets", and seem to set the scene for an analysis of ozone data which never arrives. Actually, there doesn't seem to be much of any specific introduction of the issue of offsets in this subsection.

Pg 3, l32: "but this will not. . ." If two measurement sets were both traceable to a reference, why wouldn't this fully address the challenge of merging the data sets? Is the calibration referred to here only at a single time, or could it be continuous?

Pg 3, l36: Temporal changes in sampling can also contribute to drifts.

Pg 3, l39: If drifts in ozone were up to 5%, but were statistically insignificant, then the case of ozone seems to be very different than that of solar irradiance.

Pg 5, l9: A clear definition of "jump" is needed: I assumed it to be the instantaneous addition of a constant offset, but if a jump "can last from less than a few hours to multiple years" it sounds more like a two-step process.

Pg 5, l15: Why is the requirement for a long-term stable record difficult to justify?

Pg 10, l7: "better behaved" is not very helpful: this sentence doesn't explain what monthly data is better than.

Pg 10, l12: Given that the example below gives a case in which "1.96" is not the valid multiplier, it would seem appropriate to replace "1.96" in equation 2 with a placeholder variable for the student-t distribution (as a function of n). Otherwise, a quick reader may overlook the fact that 1.96 holds only for large n.

Pg 11, l8: "we can increase our measurements per month" in this example case, but not in all circumstances. I think the point is that with enough measurements, the random measurement errors in the mean are small enough to ignore, the only source of variance is the natural variability.

Pg 13: l1: A short derivation of Eq. 4 would be useful here if possible, otherwise the term introduced to account for the jump is not intuitive.

Pg 13: l6-8: These sentences talk about fitting of the offset and drift, simultaneously and sequentially. However, to this point there has been no discussion of "fitting", only using the equations to estimate the length of time needed to estimate an offset or drift. How dos the concept of fitting, simultaneously and sequentially, affect the use of Equations 2-4?

Pg 16, l4: Is a reference really needed to support the statement that "Earth observations often invoke spatial and temporal variations"?

Pg16, l17: An error in the drift which is half the trend one is seeking to detect seems large: does it mean that in a worse-case, the detection of the trend might take twice as long (as the case with no drift in the measurement?). How was this threshold decided?

Pg 16, l42: Given only Fig 3, the optimal is obviously as many years as possible. The optimality issue only is apparent when you consider the costs (which are discussed below).

Pg 17, l16: First, a subjective evaluation ("nice") of the work of Morss et al. (2005) is probably not appropriate here, and secondly, there's not much in the sentence to really inform the reader of the relevance of this work to the present study: "a case study based on primarily hypothetical valuation estimates" doesn't help much.

Editorial comments

Pg 1, l33: delete "may"

Pg 1, l 36: either "Extensions . . . are" or "Extension . . . is"

Pg 2, l2: "assess the stability" relates to identifying and quantifying drift, but much of the paper deals also with identifying and quantifying offsets, so I wonder if this first sentence of the paper should be more general.

Pg 2, l8: The "sensitivity degradation mechanisms" described in the prior sentence will firstly impact the individual satellite record, not just the merged record referred to here.

Pg 2: l28: "Should an offset. . .", This sentence joins two statements with a semicolon, but it's not clear how or why the two statements are linked.

Pg 3, l43: why "potential"?

Pg 5, ll35-40: Some pretty general material here which seems repetitive to the introduction.

Pg 6, ll9-10: Intra-sentence repetition.

Pg 6, l23: deploy->deployment

Pg 6, l26: "Here we are. . ." has been explained before.

Pg 10, l10: Here and elsewhere, "years" are discussed in the text, while the equation is written in terms of months.

Pg 10, l36: why brackets around "%"?

Pg 10, l47: The sentence which includes the equation seems to not quite make sense.

Pg 11, l2: do you not specify the drift, rather than "estimate" it?

Tegtmeier, S., Hegglin, M. I., Anderson, J., Bourassa, A., Brohede, S., Degenstein, D., Froidevaux, L., Fuller, R., Funke, B., Gille, J., Jones, A., Kasai, Y., Krüger, K., Kyrölä, E., Lingenfelser, G., Lumpe, J., Nardi, B., Neu, J., Pendlebury, D., Remsberg, E., Rozanov, A., Smith, L., Toohey, M., Urban, J., von Clarmann, T., Walker, K. A. and Wang, R. H. J.: SPARC Data Initiative: A comparison of ozone climatologies from international satellite limb sounders, J. Geophys. Res. Atmos., 118(21), 12,229-12,247, doi:10.1002/2013JD019877, 2013.

―――――――――――――――

---

## Referee Comment (RC2) · Anonymous Referee #2 · 2 Mar 2017

General comments

This manuscript examines some challenges affecting the creation and error assessment of long-term climate-related data records, and provides some guidance in terms of the number of overlapping years between two satellite measurement systems for a given overall stability verification goal. The authors are generally well-versed in the subject matter and have useful insights to offer in this domain. The methodology is sound, especially as they make it clear that there is no "one size fits all" approach either, with so many possible variables in terms of potential instrument issues (jumps, offsets, drifts,...) as well as geophysical parameters and variations (both spatial and temporal). In this sense, given the somewhat limited case that is discussed (as this is

most directly relevant to solar irradiance type data), the results should be viewed really as applied to these very specific conditions. Would the authors agree/disagree or care to comment further (see also the next comment)?

However, I would argue that the example used is somewhat limited given the broader interest from the atmospheric community in terms of atmospheric data (temperatures as well as composition), and the larger range of variabilities (spatial and temporal) tied to such observations. Can anything useful not be mentioned in this regard, even for a somewhat specific case (such as ozone or water vapor data) using two "state-of-the art" monthly zonal mean data sets? Is the feeling that the overlap period needs would be longer, shorter, or about the same, even if this is based on a semi-quantitative approximate analysis (or why can this not be "readily added"?). I suspect it is in fact somewhat complicated and (clearly) quite dependent on the data sets themselves, but it would be good to add some words to this effect, if at all possible, even if a "full-blown" analysis is not performed for this other case of even broader interest. The value of this work is limited because of the limited example, although this is clearly useful to the solar irradiance community.

I also believe that one should never lose track of the fact that even if one can consider it valid to require (as a minimum) two years of overlap between satellite measurements in order to verify and "guard against" (if possible) potential relative drifts or jumps, the overall (longer-term) satellite record still depends very much on the years prior to and following the overlap period, for which independent observations (or reference calibration) are still required to check stability over the long run. In other words, current stability does not guarantee future stability (or even past stability). The authors might consider adding such additional words of caution, although there are generally enough cautionary sentences in the manuscript that this criticism is a fairly soft one; possibly this is more a question of separating the cautionary portions in a slightly cleaner way, as there is some overall inefficiency in the presentation and clarity (too many words or sentences in some places).

Overall, I would recommend publication of this work as useful knowledge for readers interested in careful planning and evaluation of long-term observational (and merged) data records. Whether this manuscript is appropriate enough for ACP, or possibly better suited for a different Journal, is something that the editors should probably ponder further, but I would not be strongly opposed to this manuscript finding a place in ACP (although AMT may be a better fit), after some further consideration of the reviewer comments (mostly the general comments). I also do agree broadly with the other review of this manuscript, submitted shortly before this one.

Specific comments (mostly minor comments for added clarification and a number of editorial details)

Page 1, Line 19 (P1-L1): maybe change slightly to "Satellite sensors can enable unprecedented understanding of the Earth's climate system by providing measurements of incoming solar radiation, as well as both passive and active...and temporal coverage." [the last part of this long sentence seems superfluous - this is clearly why satellites are outstanding in the aspects mentioned]

P1-L33: delete "may" before "occur".

P1-L36: change "are" to "is".

P2-L30: This paper presents techniques that can address...

P2-L44: "showed that ..."

P3-L43: While this manuscript focuses mostly on solar irradiance records, there are some selected refs. to other parameters and ozone, for example. For the more general reader, adding at least a WMO (2014) reference for such studies seems appropriate here (at least) [and maybe also "and references therein", given the non-negligible amount of work that has gone into merging various data records of atmospheric temperature and composition]. A more careful consideration of various other refs. could be considered, although this is not central to this manuscript's relevance, given what is

discussed already with a few other refs mentioned.

P3-L49: I am not sure that polar ozone loss is generally viewed as such a "major environmental concern" for the decades ahead [but if the consensus exists among this small subset, no major problem in keeping such strong words I suppose].

P4-L26: probably requires the word "is" at the end of the sentence.

P4-L39: Another ref. worth mentioning, it would seem, is the Hubert et al. [AMT, 2016] work on relative drifts between a large number of ozone time series measurements, somewhere in this paragraph, for example. ["and references therein..." at the very least].

P4-L44: delete "same" before "problems".

P5-L3: add "this" before "is often beyond".

P5-L7: replace "efforts" by "effects" (?)

P5-L23: As mentioned in my general comments, the "final data products" cannot just depend on the overlap record... A "best check", maybe, but not if the overlap period is so short compared to the full record time period (and some records extend for more than a decade).

P5-L36: "are applicable to" could be reworded as "may be useful for".

P6-L25: "such criteria are impractical".

P7-L6: "contains all three uncertainty sources identified in this study."

P8-L27: I suggest deleting "as is".

P9-L3: change "difference" to "differences". Also on line 8.

P10-L12: To get to equation (2) from eq. (1), you should specify where the 1.96 comes from, as this is not just a simple inversion of (1). The text in the paragraph that follows carries some of the information needed, but a more crisp description (or sentence

before equation (2)) would be much better.

P10-L29: "not appropriate"? Clarify what this means here please.

P10-L32: "users should never ignore the added uncertainty" - not very clear what this means here; is the "added uncertainty" not basically built into the merged datasets as a result of the merging process, so either the variability changes somewhere or there are small steps/jumps, or "built-in" trends that differ from any individual dataset to some extent. But when one looks into the trends using the merged dataset, one gets a result that carries an uncertainty, e.g. by using formulae or methodologies suggested by the authors (or other past work). Do you mean, following the text until the end of this section 3, that users should try their best to "reduce the uncertainty in the overlap adjustments"? This is also something that is often a "given", with little recourse besides independent confirmation using another dataset, although there are cases where some other methods might be used, as the authors mention. Maybe one just needs to say something like "users should be aware of added uncertainties…" and "try to minimize such effects, whenever possible" [and give one or two specific examples where this can be done].

P10-L48: If I use a drift factor in equation (3) that changes by a factor of 2, the resulting ratio for number of months would seem to go as (1/2) to the power 2/3, or a factor of 0.63. If I use 0.63 times 3 years, I get 1.9 years, which is close to the ratio in Fig. 3, but not exactly a factor of 2./3. (0.67). Could you clarify this further or point to equation (3) specifically when discussing Fig. 3 to give the reader a clearer explanation for how the curve changes. One needs 5 years or so to get another factor of two drop in the drift value (or drift error reduction).

P13-L7: delete "the" before "a drift".

P14-L15: change "produces" to "produce", as "the jumps" refers to the subject; also I suggest changing "on" to "of" before "Fig. 4".

[Figure]

P14-L30: add a space before "represented".

P16-L8 to L11: Not only can ground-based and in situ observations help during the overlap period, but they can also help by checking the longer-term trends including those for the merged datasets; this is actually a chance to give a nice "plug" for the usefulness of such independent observations rather than just limiting them to an overlap period, in my view - not that the intent was to limit their usefulness, but why not be more inclusive?

P16-L12: "efforts have been put forth" sounds better (for example).

P16-L37: Delete the period after "standards".

P16-L40/41: I would shorten the English here, e.g. "Figure 3 shows that drift detection accuracy improves as a function of the number of overlap years. Improvements in drift detection capability decrease as the number of overlap years increases, but the optimal overlap duration is difficult to identify."

P16-46: Instead of "may be proposed to improve", I would say "have been proposed and implemented..." [this is the case].

P17-L1-L23: I found this discussion somewhat long, although it is not that long compared to the full text - at least it is of less interest to some readers for a publication intended for atmospheric chemistry and physics (with solar radiation included).

P17-L18: I would add a comma after "satellite data".

P17-L29 to L34: Again here, I would prefer to see this as an opportunity to present ground-based observations as a cost-effective alternative, or certainly a complement, to satellite data (for certain applications, at least). The "potentially high cost" (line 34) comes more from the satellite observations, in comparison to ground-based networks (for atmospheric composition applications in particular).

P18-L10: "achieved due to drift ..." needs some improvement, maybe "achieved by

considering that drift is inversely proportional to the number. . .".

P18-L17: Again, I would add something here regarding ground-based, not to consider satellites as a final word for everything. . . e.g. "as observational approaches, both from satellites and from the ground,are considered. . ." This is true even if your main goal here is to provide some quantifiable basis for satellite overlap goals and methodologies, regardless of the additional input potential from non-satellite data.

Appendix A: This one could be summarized in a few sentences since the results of monthly versus 27-day period averages etc. . . are so similar for both cases. If you insist on keeping it, it could be dramatically shortened (which is why just adding a sentence or two in the main text seems appropriate to me - if at all), there are minor typos. L5, change "word" to "world", L6, delete "the" before "June", and the solar SIM std. dev. number needs an exponential format type (superscript "4") like the other numbers.

Appendix B: On L10, the "with" does not need to be capitalized. More importantly, the Fig. captions should describe the Fig. contents, y and x axes (unless it is plainly obvious), and not add lines of comments that really belong in the text/discussion (and might be duplicated unnecessarily). In Fig. B2, the caption should also mention what the dashed blue lines represent instead of mentioning what this Figure "supports" (do this in the text). For Fig. B3, in addition, the "theoretical quantities" are too much of a mystery, even with refs. that the reader could check out, so adding a sentence or two to actually describe what is being plotted would be very beneficial, or give examples if there are really too many different quantities. . .

Appendix C: L6, please correct the reference to the plot and Figure number (no third plot in Fig. 2); also, using the word "visually" seems unnecessary. On line 21, I would change "the" to "a" before "signal-to-noise", and add "that" before "is slightly". On L27, I would delete "we can imagine that the detection of" and just say "this drift,. . .", and L29, I would add "it" before "would very". On Fig. C1 caption, delete "the" before "even small drifts", on L4, add "the" before "width", and on L5, change "time" to "times", and add I

suggest "and the probability is indicated...". On L11, change "directed" to "detected". L13-14, I suggest "of detection with more years of overlap." On the question of scale for the shaded/striped blue or green areas, this scaling is arbitrary it seems, not something that can be checked against the y-axis scale; is the maximum close to 100%? It may be worth clarifying this scale range as the y-axis does not help (or one could add a separate axis on the right side of the plot).

Appendix D: this one is pretty nebulous for those not close enough to statistics from the economics side (most readers of ACP or AMT), and I suppose this could be shortened a bit as well, especially as it is not the main goal of the manuscript (which is why after all, this is in an Appendix). Also, it sounds quite vague given the comments on lines 33-34, and/as the actual specifics are tough to provide reliably for the case at hand, let alone the (little mentioned) range of possible applications in the atmospheric sounding domain. Minor detail, footnote (4), there is a closing parenthesis missing after "(see Morss et al., 2005...". Also, the typesetting for the mathematical quantities seems off in some cases (notably for $q$ on lines 16 and 20). On L4 of page 25, I would change "use along" to "use of" or "using", or something else to clarify this.

---

## Author Comment (AC1) · 4 May 2017

**Response to Reviewers**

We would like to sincerely thank both reviewers for helping us make this a significantly better paper. Specifically, we feel the paper has benefited by inclusion of more geophysical datasets and a re-organization that has resulted in a reduction in redundancy. We are grateful for the thought and care that both reviewers showed in their reading and comments. Specific responses are below.  Our responses are in black font.

**Anonymous Referee #1**

General comments

This paper examines the issue of quantifying differences in satellite measurements sets based on overlapping measurement periods, addressing the question of how long overlap periods need to be to accurately estimate offsets and drifts between instruments. A few general formulas are presented to calculate required overlap periods for given desired precision requirements of offset and drift estimation, and examples are presented.

While the issues discussed are definitely relevant to the construction of long-term atmospheric data records, this paper does not actually directly deal with any atmospheric measurements, instead focusing on examples based on satellite measurements of solar irradiance. As a result, the fit between this paper and ACP (or even AMT for that matter) is somewhat questionable. Readers of ACP would likely benefit greatly from examples using actual atmospheric data, and given the ready availability of multi- instrument data sets like stratospheric ozone (e.g., Tegtmeier et al., 2013), it wouldn't be hard to include such examples. Of course, atmospheric data, with the temporal and spatial variability that comes with it, may present some additional complications to the analysis (which is mentioned in passing in the manuscript), but a discussion of these complications seems warranted in such a paper if it truly wants to address the analysis (and merging) of atmospheric data.

We appreciate this comment and in response have included two additional atmospheric datasets (ozone and temperature) to help illustrate the problems in Earth observations from satellite and the importance of these techniques for estimating appropriate satellite overlap. We show how atmospheric data have more complications than solar irradiance data for a variety of reasons including spatial and temporal matchup as well as differences with latitude and season. We have also added more clarity on the importance of solar data to understanding atmospheric behavior, including variability, climatology and change.

The utility of this paper to atmospheric community could also be improved by a fuller description of the general implications of the analysis before descending to the focused solar irradiance example. For example, Fig 3 displays the detectable drifts in the solar irradiance data sets as a function of years of overlap, which suggests a general form of

the solution, but won't provide any quantitative information to anyone working with other data sets. Instead, a plot of ratio of drift to variability as a function of n (perhaps for different sample values of autocorrelation) would be directly relevant to users of other data sets.

We have now highlighted the general applicability of this analysis. We have also changed Plot 3 (now Plot 5), to include results for various levels of variability and autocorrelation. This is in direct response to the reviewer's suggestion and we feel that it improves the paper. While we assume anyone making use of this paper will make use of the formulae not the figures, we feel that Plot 5 is now more useful in helping users understand the implication of the formulae.

Not written in this paper, although I think the two reviewers seem to have understood this point, is that we would like decisions about pre-flight calibration, satellite overlap, etc. to be based on scientific criteria and not on bureaucratic guesses as to what level of overlap is adequate. We do not feel this paper has completely answered the question of overlap, but we do hope we have helped start a series of scientific discussions on the subject.

Stylistically, I found the paper repetitious in places, often returning to discussions of issues that aren't, in my opinion, of central importance. For example, the issue of requirements (or the desire) for self-calibrating, consistent systems for atmospheric measurements is often brought up, but this paper deals specifically with techniques to deal with situations where measurements are not self-calibrated. This point can be made succintly in the introduction, and thereafter neglected, at least until the discussion and conclusions.

We have taken these comments to heart and agree with the reviewer. In response, we have brought all discussions of self-calibrating satellites into one place. Similarly, we brought discussions on the complications of Earth observations (spatial match-up, etc.) into one place. We have merged the discussion of what had previously done on offsets, drifts and jumps that occurred in the rather long introduction into the body of the paper where we deal with offsets, drifts and jumps. In doing this merging we were able to eliminate many redundancies. Again, we sincerely thank the reviewers for pointing this out. The paper is greatly improved by these comments.

Specific comments:

Pg 1, l27: "offset or a drift in the offsets": If "offset" is singular in the first case, then "drift in the offset" seems more appropriate. But the sentence was confusing to me at first, and I wonder if just "drift" is easier to understand.

We have made the change and tried to make the point more clear.

Pg 1, l37-38: "may also benefit. . ." this issue is not dealt with in any substantial way in the paper, so this statement's inclusion in the abstract seems superfluous.

Thank you. This is dealt with in the re-write of our paper by including examples of ozone and temperature in Section 2 of the revised manuscript.

Pg 2, l16: "tying data to absolute reference standards with the intent of developing traceability to reference standards" sounds a little tautological.

We agree. This wording has been completely redone—see the new wording in the response to the next point.

Pg 2, l17: It's not clear to me why reference standards are brought into the argument here, is the point that if one of the two overlapping measurement sets is a standard, then you can extend a standard through identification of an offset and drift in the second instrument?

Thank you. We have hoped to make the point more clear with the following:

Page 2, Line 35: *"Another approach to addressing satellite uncertainty, based on maintaining traceability through on-board calibration capabilities using absolute references, has been advocated through the CLARREO and TRUTHS programs (Wielicki et al., 2013; Fox et al., 2013). For both programs, verification of merging of these new approaches will be important for validation of expected agreement. "*

Pg 2, l36: Do "wavelength scale corrections" etc. really help instrument scientists understand the fundamental observations? Or does an understanding of the fundamental observations allow for valid corrections?

Thank you. We think the wavelength scale corrections truly are helpful—and nearly universally examined by satellite instrument scientists. We need to understand these details of the instrument (including those related to wavelength scale corrections) before we can understand the observation. All of these effects have a different influence on the data. All of these components are terms in the measurement equation, and if there is something 'missing' in the measurement equation then it is possible for the observation to be misinterpreted. We have not modified this sentence in the text, but hope the new context makes the point more clear. The new wording starts on Page 2, around line 32.

Pg 3, l15: Does removing a bias affect the precision of the merged data set? And, does one really need to remove a bias to identify a drift? If you look at changes with time (time derivatives) the absolute value doesn't matter.

This is part of a re-written paragraph. We believe that the answer to the question: 'Does removing a bias affect the precision of the merged data set?' is yes it does. If left in place in a merged data set, a bias in one component of a merged dataset can be misinterpreted as drift. This situation has much in common with our discussion on jumps – see the discussion regarding Figure 6 in our paper. The ability to use time derivatives is strongly dependent on the signal-to-noise ratio of the measurement and the autocorrelation of the data. Our revised Figure 5 helps illustrate this matter. We hope the new text is clearer.

Pg 3, l28: It would seem that the paper is of interest to a wider group than just the users of merged data sets, specifically to the creators of merged data sets.

Thank you. We have adjusted the paper so that users of merged data sets may be able to more fully appreciate the uncertainties add to the data—uncertainties that aren't always fully explained when one is interested in downloading a dataset. The addition of creators as an important group is now added on Page 3, Line 19-20.

Pg 3, ll40: The first two paragraphs of Sec 1.1 have no apparent specific connection to "Offsets", and seem to set the scene for an analysis of ozone data which never arrives. Actually, there doesn't seem to be much of any specific introduction of the issue of offsets in this subsection.

Thank you. I believe this is addressed in the larger re-write of the manuscript.

Pg 3, l32: "but this will not. . ." If two measurement sets were both traceable to a reference, why wouldn't this fully address the challenge of merging the data sets? Is the calibration referred to here only at a single time, or could it be continuous?

Removed in revised manuscript. The point we still want to make is for the case when only one measurement is traceable. I think the re-write addresses this on See Page 2, line 21. I believe it is now clearer. However, to the reviewer's point of two traceable datasets should make merging of datasets quite simple, I would still say that challenges will exist. For instance, if each of two temperature satellites are traceable with an uncertainty on the temperature of +/1 1 degree, we could easily have an offset between the two satellites of 2 degrees. If the uncertainty on the traceable products is quite, quite small, then I agree with the reviewer, although we still probably want to verify.

Pg 3, l36: Temporal changes in sampling can also contribute to drifts. C3

Thank you. We agree. We now include this on page 3 starting at line 12.

Pg 3, l39: If drifts in ozone were up to 5%, but were statistically insignificant, then the case of ozone seems to be very different than that of solar irradiance.

 Indeed, the ozone and irradiance cases are very different, but the kind of analysis we perform here is applicable regardless of the case under study. The climate data record mandates different levels of knowledge for these two cases. We address this on Page 6 starting at line 13.

Pg 5, l9: A clear definition of "jump" is needed: I assumed it to be the instantaneous addition of a constant offset, but if a jump "can last from less than a few hours to multiple years" it sounds more like a two-step process.

Thank you. We now write: *"Jumps are permanent or semi-permanent level shifts in the data that occur at specific points in time and are not attributable to the parameter being observed; jumps could represent a change in sensitivity of an instrument or a change in location or orientation of the satellite."*

Pg 5, l15: Why is the requirement for a long-term stable record difficult to justify?

Context changed in the formulation of section 1.2 'Planning for need homogeneity'. Some of the difficulty comes from the concept that we don't want to require stability that is unattainable. There can be a bit of a dance between what we want and what is possible. Another reason for the difficulty is that there are many, many uses of any environmental dataset, so the requirement for one application may be different from the requirement (on the same dataset) for another application. It may be tempting to take the smallest requirement from any user community, but strict stability can be hard to achieve and very expensive.

Pg 10, l7: "better behaved" is not very helpful: this sentence doesn't explain what monthly data is better than.

The sentence has been rewritten to avoid the unhelpful and unclear wording. New text is on Page 12, line 4: *"These constraints for the formula are some of the reasons that monthly averages are used as often as they are: monthly averaged data remove higher frequency noise and sampling match-up problems from different instruments are minimized."*

Pg 10, l12: Given that the example below gives a case in which "1.96" is not the valid multiplier, it would seem appropriate to replace "1.96" in equation 2 with a placeholder variable for the student-t distribution (as a function of n). Otherwise, a quick reader may overlook the fact that 1.96 holds only for large n.

We have rewritten this section and believe the issues are now addressed appropriately on page 12, line 35. We worry that removing the 1.96 from the formula would make it less useful and intuitive. In reality, most applications will be in the large n limit where 1.96 holds or is at least close.

Pg 11, l8: "we can increase our measurements per month" in this example case, but not in all circumstances. I think the point is that with enough measurements, the random measurement errors in the mean are small enough to ignore, the only source of variance is the natural variability.

Small changes in the text have been made to clarify this point. I'm not sure that it is safe to write that "the only source of variance is the natural variability." If this were true, good observations would match up perfectly in terms of variability, but they don't for reasons now more clearly itemized in the text.

Pg 13: l1: A short derivation of Eq. 4 would be useful here if possible, otherwise the term introduced to account for the jump is not intuitive.

We have added a new equation that shows how Eq. 4 is derived. The added equation and associated sentences address this concern—those interested can derive what is now equation (5) from given equation (4), which is the new addition. This work has already been published in Weatherhead et al., 1998.

Pg 13: l6-8: These sentences talk about fitting of the offset and drift, simultaneously and

sequentially. However, to this point there has been no discussion of "fitting", only using the equations to estimate the length of time needed to estimate an offset or drift. How dos the concept of fitting, simultaneously and sequentially, affect the use of Equations 2-4?

Thank you. We've adjusted the text so that we talk about the future fitting of the data. We hope this is more clear. Fundamentally, how long we need to monitor depends on how we will fit to an offset and a drift term in the data. We now write in terms of, "...*If we assume that we are going to fit the environmental data to a linear statistical model of the form:...*" We hope this is now clear.

Pg 16, l4: Is a reference really needed to support the statement that "Earth observations often invoke spatial and temporal variations"?

Statement lost in rewrite.

Pg16, l17: An error in the drift which is half the trend one is seeking to detect seems large: does it mean that in a worse-case, the detection of the trend might take twice as long (as the case with no drift in the measurement?). How was this threshold decided?

This is being introduced in this paper after multiple discussions among co-authors who correctly pointed out that for many parameters there is not requirement for an error in the drift. WMO, under the guidance of Global Climate Observing System, is working on establishing these requirements, but the choices of drift limit are, in most cases quite arbitrary. We agree that this drift is large, but it is also quite likely that this large of a drift (half the size of the trend) is not attainable for many important observations. Evaluation of stability and limitations of drift for various parameters must be done very, very carefully and is therefore beyond the scope of this paper.

Pg 16, l42: Given only Fig 3, the optimal is obviously as many years as possible. The optimality issue only is apparent when you consider the costs (which are discussed below).

This section is reworded in a very similar way. Please see page 20, line 31.

Pg 17, l16: First, a subjective evaluation ("nice") of the work of Morss et al. (2005) is probably not appropriate here, and secondly, there's not much in the sentence to really inform the reader of the relevance of this work to the present study: "a case study based on primarily hypothetical valuation estimates" doesn't help much.

On page 21, line 5 the sentence now reads:

Morss et al. (2005) provide an overview of relevant economic concepts and theory for optimal design of observational systems based on benefit cost tradeoffs.

Editorial comments Pg 1, l33: delete "may" Pg 1, l 36: either "Extensions . . . are" or "Extension . . . is"

done

Pg 2, l2: "assess the stability" relates to identifying and quantifying drift, but much of the paper deals also with identifying and quantifying offsets, so I wonder if this first sentence of the paper should be more general.

Thank you.  It now reads, "…assess the characteristics.."

Pg 2, l8: The "sensitivity degradation mechanisms" described in the prior sentence will firstly impact the individual satellite record, not just the merged record referred to here.

We agree with the reviewer on this point. For an individual satellite record, analysis of the degradation measurement equation must be evaluated. Data are adjusted for all known factors. But only through the process of comparing simultaneous data records (i.e.overlaping) can any potential systematic errors be evaluated. We have not edited this comment since the purpose of the paper is to discuss the merging of records, but we think the new context makes it more clear.

Pg 2: l28: "Should an offset. . .", This sentence joins two statements with a semicolon, but it's not clear how or why the two statements are linked.

Sentence rewritten

Pg 3, l43: why "potential"?

'potential' removed

Pg 5, ll35-40: Some pretty general material here which seems repetitive to the introduction.

We believe we have cut down significantly on the redundancy.  The new sentence reads:

"The techniques discussed herein are applicable to instrument scientists pursuing improvements in on-board instrument corrections, but also for mission planning by program managers to ensure the best overlap characteristics of adjoining missions."

Pg 6, ll9-10: Intra-sentence repetition.

On page 7, line 40, the sentence now reads:

*The instruments for the SORCE mission are described in a series of papers published in Solar Physics related to the design, operation, calibration, and performance of the SORCE instruments. Harder et al. (2005a) describes the scientific requirements, design, and operation modes for the instrument.*

Pg 6, l23: deploy->deployment                                          Pg 6, l26: Here w

done

Pg 10, l10: Here and elsewhere, "years" are discussed in the text, while the equation is

written in terms of months.

Cleared up in text modifications.  We now present the equations as years and the words are more often in "length of time" or "amount of time."  We have only left "months" when we are referring to fractions of a year, or think it is more clear

Pg 10, l36: why brackets around "%"? Pg 10, l47: The sentence which includes the equation seems to not quite make sense. Pg 11, l2: do you not specify the drift, rather than "estimate" it?

Done.

Again, thank you to the reviewer.

---

## Author Comment (AC2) · 4 May 2017

**Response to Reviewers**

We would like to sincerely thank both reviewers for helping us make this a significantly better paper. Specifically, we feel the paper has benefited by inclusion of more geophysical datasets and a re-organization that has resulted in a reduction in redundancy. We are grateful for the thought and care that both reviewers showed in their reading and comments. Specific responses are below. Our responses are in black font. Quotes from the new draft are in italics.

**Anonymous Referee #2**

General comments

This manuscript examines some challenges affecting the creation and error assess-ment of long-term climate-related data records, and provides some guidance in terms of the number of overlapping years between two satellite measurement systems for a given overall stability verification goal. The authors are generally well-versed in the subject matter and have useful insights to offer in this domain. The methodology is sound, especially as they make it clear that there is no "one size fits all" approach either, with so many possible variables in terms of potential instrument issues (jumps, offsets, drifts,...) as well as geophysical parameters and variations (both spatial and temporal). In this sense, given the somewhat limited case that is discussed (as this is most directly relevant to solar irradiance type data), the results should be viewed really as applied to these very specific conditions. Would the authors agree/disagree or care to comment further (see also the next comment)?

Thank you. We believe we have now made the paper much more relevant to the atmospheric science community by including examples from ozone and temperature and by expanding Figure 3 (now Figure 5) to show how results would differ with other values of variability and autocorrelation. We have also focused the writing on the applications to satellites monitoring atmospheric parameters and the associated problems. We appreciate the challenge to make this paper more generally useful to the large community interested in atmospheric observations. While the figures are devoted to application of the proposed techniques, the formulae presented are what we feel are the true contributions. As such, the formulae are quite generally applicable to the broad atmospheric community. We would also like to point out that approximately half of the co-authors involved in this study are focused on non-solar satellite observations.

However, I would argue that the example used is somewhat limited given the broader interest from the atmospheric community in terms of atmospheric data (temperatures as well as composition), and the larger range of variabilities (spatial and temporal) tied to such observations. Can anything useful not be mentioned in this regard, even for a somewhat specific case (such as ozone or water vapor data) using two "state-of-the art" monthly zonal mean data sets? Is the feeling that the overlap period needs would be longer, shorter, or about the same, even if this is based on a semi-quantitative approximate analysis (or why can this not be "readily added"?). I suspect it is in fact somewhat complicated and (clearly) quite dependent on the data sets themselves, but it would be good to add some words to this effect, if at all possible, even if a "full-blown" analysis is not performed for this other case of even broader interest. The value of this work is limited because of the limited example, although this is clearly useful to the solar irradiance community.

Thank you. We added examples from for overlapping satellites of ozone (New Figure 1 and associated text) and temperature (New Figure 2 and associated text) for illustration purposes. We had intended to get appropriate ozone and temperature data to carry out the sample calculations on. Indeed, with a number of co-authors on this paper, we thought it would be do-able. We finally concluded that it was beyond the scope of this paper to apply the techniques fully to temperature and ozone. We hope the reviewer and editor will see our efforts to be more inclusive of other datasets as a direct response the reviewers' helpful comments.  We also do view solar radiation as a very important parameter for understanding the Earth's chemistry and climate.

I also believe that one should never lose track of the fact that even if one can consider it valid to require (as a minimum) two years of overlap between satellite measurements in order to verify and "guard against" (if possible) potential relative drifts or jumps, the overall (longer-term) satellite record still depends very much on the years prior to and following the overlap period, for which independent observations (or reference calibration) are still required to check stability over the long run. In other words, current stability does not guarantee future stability (or even past stability). The authors might consider adding such additional words of caution, although there are generally enough cautionary sentences in the manuscript that this criticism is a fairly soft one; possibly this is more a question of separating the cautionary portions in a slightly cleaner way, as there is some overall inefficiency in the presentation and clarity (too many words or sentences in some places).

We fully agree with the reviewer on this point. He/she is correct that we can not forget that the stability of an instrument during overlap is not evidence of how it behaves

outside of the overlap period. We have tried to emphasize this point by gathering in one place many of our caveats on the limitations of information in an overlap period. (See page 3, line 23) We don't want to oversell this point, though, out of concern that an inexperienced reader could think that overlaps in satellites are not very important. By pulling together our thoughts more cogently in addressing the value of the overlap period, we hope we have addressed this point.

Overall, I would recommend publication of this work as useful knowledge for readers interested in careful planning and evaluation of long-term observational (and merged) data records. Whether this manuscript is appropriate enough for ACP, or possibly better suited for a different Journal, is something that the editors should probably ponder further, but I would not be strongly opposed to this manuscript finding a place in ACP (although AMT may be a better fit), after some further consideration of the reviewer comments (mostly the general comments). I also do agree broadly with the other review of this manuscript, submitted shortly before this one.

Thank you. We believe the revised manuscript appropriately addresses a number of atmospheric data issues and will be happy if the paper stays in ACP. We also recognize that ACP has a history of publishing papers on solar data, so think this also makes a good fit. We have added sentences to reinforce the importance of solar data to atmospheric sciences.

Specific comments (mostly minor comments for added clarification and a number of editorial details)

Page 1, Line 19 (P1-L1): maybe change slightly to "Satellite sensors can enable un-precedented understanding of the Earth's climate system by providing measurements of incoming solar radiation, as well as both passive and active...and temporal coverage." [the last part of this long sentence seems superfluous - this is clearly why satellites are outstanding in the aspects mentioned]

We happily shortened this sentence in response to both your comments and the other comment we received. Thank you.

P1-L33: delete "may" before "occur".

Done

P1-L36: change "are" to "is".

Thank you.

P2-L30: This paper presents techniques that can address...

Much better. Thank you.

P2-L44: "showed that ..."

Thank you. .

P3-L43: While this manuscript focuses mostly on solar irradiance records, there are some selected refs. to other parameters and ozone, for example. For the more general reader, adding at least a WMO (2014) reference for such studies seems appropri- ate here (at least) [and maybe also "and references therein", given the non-negligible amount of work that has gone into merging various data records of atmospheric tem- perature and composition]. A more careful consideration of various other refs. could be considered, although this is not central to this manuscript's relevance, given what is

Thank you. We've made the addition and corrections.

C3discussed already with a few other refs mentioned.

P3-L49: I am not sure that polar ozone loss is generally viewed as such a "major environmental concern" for the decades ahead [but if the consensus exists among this small subset, no major problem in keeping such strong words I suppose].

Polar ozone loss ("The Antarctic Ozone Hole" and "Arctic Ozone holes") remains to be an important benchmark for environmental sciences. The entire process from identifying the problem, confirming the cause, communicating with decision makers and reversing the concentration of ozone depleting substances is a benchmark of the important role of environmental science in society. The final success will be the cessation of polar ozone holes, which may not occur for a few decades. If we don't see this full recovery, there may be reason to doubt the highly successful collaboration of scientists and policy makers. So, if the reviewers have no objections, we'd like to let this statement stand.

P4-L26: probably requires the word "is" at the end of the sentence.

Thank you. Done.

P4-L39: Another ref. worth mentioning, it would seem, is the Hubert et al. [AMT, 2016] work on relative drifts between a large number of ozone time series measurements,

somewhere in this paragraph, for example. ["and references therein..." at the very least].

Thank you.  Done. And, particularly more appropriate now that we have added the ozone example.

P4-L44: delete "same" before "problems".

Thank you. Done.

P5-L3: add "this" before "is often beyond".

Thank you. Done.

P5-L7: replace "efforts" by "effects" (?)

Thank you. That sentence has been deleted due to our reorganization and reduction of redundancies.

P5-L23: As mentioned in my general comments, the "final data products" cannot just depend on the overlap record... A "best check", maybe, but not if the overlap period is so short compared to the full record time period (and some records extend for more than a decade).

The reviewer is right, of course; we agree fully. We have reworded this point to make it more clear in several places. We modify the cited line with, "*In Section 8 we offer an approach to evaluate how important overlap is compared to other choices that can help improve a long-term data record.*"

P5-L36: "are applicable to" could be reworded as "may be useful for".

Thank you. We have adjusted this sentence, and simultaneously responded to another suggestion that we note that this can be useful to those using the data.

P6-L25: "such criteria are impractical".

Thank you. Done.

P7-L6: "contains all three uncertainty sources identified in this study."

Not sure what to do here.

P8-L27: I suggest deleting "as is".

Done. Thank you.

P9-L3: change "difference" to "differences". Also on line 8.

Thank you. Done.

P10-L12: To get to equation (2) from eq. (1), you should specify where the 1.96 comes from, as this is not just a simple inversion of (1). The text in the paragraph that follows carries some of the information needed, but a more crisp description (or sentence C4 before equation (2)) would be much better.

We have added text to explain the 1.96 factor: *"The factor of 1.96 is to support a 95% confident limit on the offset; if more confidence is needed in the offset, a higher factor can be used based on classic statistical tables."* We think this is more crisp and, along with the other changes in this section, helpful to the paper. Thank you.

P10-L29: "not appropriate"? Clarify what this means here please.

We have adjusted the text to be more clear about the student-t assumptions: *"…because the formula offers an estimate of length of time needed to limit uncertainty in an offset, and such an estimate is rarely precise to many significant digits. We conclude for the datasets we have been exposed to that after roughly two years of data collection the large number limit of 1.96, may be considered appropriate."* To be very explicit in response to the reviewer (in case this new text is not sufficient), when we estimate that it will take, for instance, three years to reduce the uncertainty on the offset to a specified level, the specific variability observed may result in the uncertainty on the offset to reach the specific level a few months earlier, or a few months later. Thus, coming to a conclusion that the overlap needs to last for 3.0179 years is not really relevant (appropriate).

P10-L32: "users should never ignore the added uncertainty" - not very clear what this means here; is the "added uncertainty" not basically built into the merged datasets as a result of the merging process, so either the variability changes somewhere or there are small steps/jumps, or "built-in" trends that differ from any individual dataset to some extent. But when one looks into the trends using the merged dataset, one gets a result that carries an uncertainty, e.g. by using formulae or methodologies suggested by the authors (or other past work). Do you mean, following the text until the end of this section 3, that users should try their best to "reduce the uncertainty in the overlap adjustments"? This is also something that is often a "given", with little recourse besides independent confirmation using another dataset, although there are cases where some other methods

might be used, as the authors mention. Maybe one just needs to say something like "users should be aware of added uncertainties. . ." and "try to minimize such effects, whenever possible" [and give one or two specific examples where this can be done].

We have added a sentence to help clarify this at page 12, line 37. The user can choose any of a number of ways to address this added uncertainty and the approaches are beyond the scope of this paper—but may make for an interesting follow up paper!

P10-L48: If I use a drift factor in equation (3) that changes by a factor of 2, the resulting ratio for number of months would seem to go as (1/2) to the power 2/3, or a factor of 0.63. If I use 0.63 times 3 years, I get 1.9 years, which is close to the ratio in Fig. 3, but not exactly a factor of 2./3. (0.67). Could you clarify this further or point to equation (3) specifically when discussing Fig. 3 to give the reader a clearer explanation for how the curve changes. One needs 5 years or so to get another factor of two drop in the drift value (or drift error reduction).

We are a little unsure of the reviewer's question. We note that the plot is a semi-log plot. We want to run through one example, as the reviewer has suggested and for ease of reading, set aside the radiation units and exponents (leaving off *10-5 watts/m2/nm/year) Using the reviewer's example and specifically using the data in Figure 5, we have a drift of 2.88  for 5 years.  If we consider twice that drift (5.76), that should take (5 years*0.63) which is 3.15 years.  That is about halfway through our datapoints for 3 years (drift detectable of 6.19) and 3.25 years (drift detectable of 5.49).  Our 5.76 estimate lies between those two values (5.49*10-5 and 6.19*10-5), so this is all making sense to us. We welcome this sort of check, but we think the data in the figure and the data in the formula agree.  We would be extremely grateful if the reviewer can point out a problem, should one exist.  At the moment, we don't see the problem.

If it would be helpful, we can add the data from Figure 3 (now Figure 5) in an appendix. Note that Figure 5 has also changed to be a larger, more informative plot. I offer the data for Figure 5, if it is helpful to the reviewer:

P13-L7: delete "the" before "a drift".

Done. Thank you.

P14-L15: change "produces" to "produce", as "the jumps" refers to the subject; also I suggest changing "on" to "of" before "Fig. 4".

Done. Thank you.

P14-L30: add a space before "represented".

Done. Thank you.

P16-L8 to L11: Not only can ground-based and in situ observations help during the overlap period, but they can also help by checking the longer-term trends including those for the merged datasets; this is actually a chance to give a nice "plug" for the usefulness of such independent observations rather than just limiting them to an over- lap period, in my view - not that the intent was to limit their usefulness, but why not be more inclusive?

Thank you. We agree and have adjusted the text (e.g. page 6, line 47)and in the conclusion to better highlight the role of independent observations.

P16-L12: "efforts have been put forth" sounds better (for example).

Thank you. That sentence has been removed due to the reorganization of text and removal of redundancies.

P16-L37: Delete the period after "standards".

Thank you, that sentence has been removed due to our re-arranging of text.

P16-L40/41: I would shorten the English here, e.g. "Figure 3 shows that drift detection accuracy improves as a function of the number of overlap years. Improvements in drift detection capability decrease as the number of overlap years increases, but the optimal overlap duration is difficult to identify."

Done. Thank you.

P16-46: Instead of "may be proposed to improve", I would say "have been proposed and implemented. . ." [this is the case].

The reviewer is right, of course, that multiple ideas have been proposed and implemented that have improved satellite records. However, we are trying to say something different in this sentence and paragraph. We want to convey that appropriate overlap of satellites is one way to help assure better data, but it is not the only way. We have adjusted the text starting on  on page 20, line 31to make that more clear.

P17-L1-L23: I found this discussion somewhat long, although it is not that long com-

pared to the full text - at least it is of less interest to some readers for a publication intended for atmospheric chemistry and physics (with solar radiation included).

We would very much like to keep this section intact, although we've modified in response to the other comments. This section may not speak to scientists, but it does speak to managers who often are the final decision makers when it comes to continued satellite overlap. We also get very strong, positive response on this aspect when we present this information in public. I believe we are seeing more environmental science – economic discussions these days and this may well be a good thing, particularly if scientists stay engaged in these cross-disciplinary discussions. We hope the editor and reviewers will indulge us here.

P17-L18: I would add a comma after "satellite data".

Done. Thank you.

P17-L29 to L34: Again here, I would prefer to see this as an opportunity to present ground-based observations as a cost-effective alternative, or certainly a complement, to satellite data (for certain applications, at least). The "potentially high cost" (line 34) comes more from the satellite observations, in comparison to ground-based networks (for atmospheric composition applications in particular).

We agree with the reviewer on this point and have added new words in the reformulated section on requirements that help clarify the undeniable value of ground-based observations, both as aids in understanding satellite records as well as on their own. Please note the changes in page 6 and in the conclusion.

P18-L10: "achieved due to drift ..." needs some improvement, maybe "achieved by considering that drift is inversely proportional to the number. . .".

Done. Thank you.

P18-L17: Again, I would add something here regarding ground-based, not to consider satellites as a final word for everything. . . e.g. "as observational approaches, both from satellites and from the ground, are considered. . ." This is true even if your main goal here is to provide some quantifiable basis for satellite overlap goals and methodologies, regardless of the additional input potential from non-satellite data.

We have done a better job of praising the value of ground based observations in the newly merged section that addresses requirements and the potential future value of inflight calibration (Page 6, line 45)and in this conclusion section. The reviewer is completely correct: satellites are not the final word for everything. We view this paper as an effort to show how to help get the best possible data out of satellites, because there are formidable issues that need to be addressed. If anything, I think the uncertainty added by overlap, as presented in this paper, can help educate our community to the idea that even if they can download a beautifully complete (temporally and geographically) satellite dataset, it is not without significant challenges.

Appendix A: This one could be summarized in a few sentences since the results of monthly versus 27-day period averages etc. . . are so similar for both cases. If you insist on keeping it, it could be dramatically shortened (which is why just adding a sentence or two in the main text seems appropriate to me - if at all), there are minor typos. L5, change "word" to "world", L6, delete "the" before "June", and the solar SIM std. dev. number needs an exponential format type (superscript "4") like the other numbers.

The question of averaging into monthly values is one that comes up relatively frequently. Sometimes scientists will discover that they get "better" results when they use daily data, or smooth the data, or some other trick. We've never written a paper on this issue, but I think the length of this section and explicit calculations will at least take a small step toward helping scientists understand that their results, if carried out correctly, will not be dependent on time period for averaging. An appendix might be just the right place for something like this—it allows the few people who are very interested in this result to see how it plays out, while not bogging down the paper for those not so interested. Thank you very much for pointing out the typos—they have been corrected

Appendix B: On L10, the "with" does not need to be capitalized. More importantly, the Fig. captions should describe the Fig. contents, y and x axes (unless it is plainly obvious), and not add lines of comments that really belong in the text/discussion (and might be duplicated unnecessarily). In Fig. B2, the caption should also mention what the dashed blue lines represent instead of mentioning what this Figure "supports" (do this in the text). For Fig. B3, in addition, the "theoretical quantities" are too much of a mystery, even with refs. that the reader could check out, so adding a sentence or two to actually describe what is being plotted would be very beneficial, or give examples if there are really too many different quantities.

We have adjusted the text so that the discussion of the Figures is in the body of the text and not in the captions. (Page 25, line 4) We have further explained how the results in Figure B2 and B3 can be interpreted. (Page 25, line 8) We hope the text is clearer, but still may be a bit confusing to those with limited background in statistics. We wanted to

include these sections to underscore that the formulae presented can be used if the assumptions behind those formulae are met. While this might not serve as a full tutorial, it at least shows that there are objective ways to decide whether one's data are behaving as an AR(1) and whether residuals are Gaussian. With the reviewer's suggestions on how to be clearer, we now feel we are doing a better job of making our point. Again, thanks to the careful reviewer!

Appendix C: L6, please correct the reference to the plot and Figure number (no third plot in Fig. 2); also, using the word "visually" seems unnecessary. On line 21, I would change "the" to "a" before "signal-to-noise", and add "that" before "is slightly". On L27, I would delete "we can imagine that the detection of" and just say "this drift,. . .", and L29, I would add "it" before "would very". On Fig. C1 caption, delete "the" before "even small drifts", on L4, add "the" before "width", and on L5, change "time" to "times", and add I suggest "and the probability is indicated. . .". On L11, change "directed" to "detected". L13-14, I suggest "of detection with more years of overlap." On the question of scale for the shaded/striped blue or green areas, this scaling is arbitrary it seems, not something that can be checked against the y-axis scale; is the maximum close to 100%? It may be worth clarifying this scale range as the y-axis does not help (or one could add a separate axis on the right side of the plot).

Thank you. We've made all of the small changes and added a reference so that the scale of the shaded areas is more clear. The new text to help indicate the scale is:*The height of the blue bar indicates the likelihood, with the linear scale being defined such that the likelihood of detection being 50% at two years; considerably higher likelihood of detection is indicated with more years of overlap.*

Appendix D: this one is pretty nebulous for those not close enough to statistics from the economics side (most readers of ACP or AMT), and I suppose this could be shortened a bit as well, especially as it is not the main goal of the manuscript (which is why after all, this is in an Appendix). Also, it sounds quite vague given the comments on lines 33-34, and/as the actual specifics are tough to provide reliably for the case at hand, let alone the (little mentioned) range of possible applications in the atmospheric sounding domain. Minor detail, footnote (4), there is a closing parenthesis missing after "(see Morss et al., 2005. . .". Also, the typesetting for the mathematical quantities seems off in some cases (notably for q on lines 16 and 20). On L4 of page 25, I would change "use along" to "use of" or "using", or something else to clarify this.

We have adjusted some of the text in the Appendix, but would like to keep this section in the final paper for two reasons. First, when we have presented this general information to

colleagues, the economic aspects are what get the most enthusiastic response. Second, if not for financial constraints, we would be able to simply have continuous overlap of all observations. The driving force for being careful about overlap is to assure that scientific use of the final data is not constrained by economic decisions to stop overlap of data too early. We hope the editor and reviewers will allow this section to remain as an appendix. The text was changed in response to all three minor details mentioned. Thank you.

Again, we thank the reviewer for such a thoughtful and likely time consuming review. We particularly thank the reviewer for double checking to assure that our calculations in Figure 5 and the Formula are in agreement.

---

## Author Response (AR2)

**Response to Reviewer**

I thank the authors for taking the suggestions of the first round of reviews into consideration.
While manuscript still does not describe actual application of the techniques to atmospheric data,
the introductory material on merging of ozone and temperature measurements does add value to
the manuscript. And the structure and readability have improved compared to the first version.

I believe however that there are some errors and inconsistencies with the formulas, calculations
and quantitative results of the study. I urge the authors to triple check their calculations,
addressing the comments listed below.

1. The factor of *12 added to Eqs 2, 3 and 5 is wrong: to convert months into years one should
divide by 12, not multiply. But I would not advocate this: since the quantities in each equation
are based on monthly means, and the value of n used to determine the scaling factor for the
conversion to a confidence interval (1.96 or other) is the number of months, I think it's best to
write these formulas a "number of months…"  *Change made.  The reviewer is right, much to my*
*chagrine.  I won't bother with explanations.  The reviewer is right.  The corrections, however,*
*were different for each of the equations.  EQ2 doesn't need any factor of 12, Equations 3 and 5*
*just need to be represented as number of months—as the reviewer suggested.  The errors were*
*introduced in our attempt to respond to the reviewer's first set of comments.*

2. Spot checking Table A1 turns up a number of errors. For example, using Eq 2 and the
numbers provided, I calculate time periods (months or solar rotation cycles) to identify an offset
of 0.0008 watts m-2 nm-1 for SOLSTICE-SIM of 5.9 and 6.6, not 5 and 6 as listed in the table.
Also, the value of 5.8 years to identify a drift of 0.0001 watts m-2 nm-1 year-1 in the
SOLSTICE-SIM timeseries is inconsistent with Figure 5, which suggests around 2.2 years.  *We*
*have redone Table 1 to be more clear on when we are using the magnitude of variability and*
*autocorrelation of the raw data and when we are using the magnitude of variability and*
*autocorrelation of the de-trended data. We have also included more significant digits to allow*
*better comparisons. Thank you.*

3. Apparently, the calculations behind Figure 5 use different values for sigma and phi than used
for the prior calculations in section 3, and used in Table A1. This is not explained, and is
therefore extremely confusing.  *Yes, thank you. we discsussed several times as co-authors*
*whether to address this directly in the text.  Some thought it was obvious, others thought it was*
*distracting.  We have now included the text: ,,"σ and φ are the magnitude of variability and*
*autocorrelation, respectively, of the differenced monthly data once any existing trend is*
*removed."  The reason behind the different sigmas and phis is that residuals of a statistical fit*
*(and their associated characteristics summarized by sigma and phi) are always different when*
*one uses a different statistical model.  We have also added more text to make clear which drift,*
*percentage drift, sigma and phi we are using in each section and updated the reference on the*
*Figure.*

4. It doesn't make sense to me why the "drift" in Eq 3 (and 5) should be used in units of yr-1,
when the sigma and phi are based on monthly timeseries. Perhaps this explains why different
sigma and phi values are used in the construction of Fig 5, these could be estimates of the standard deviation and autocorrelation of the annual mean timeseries, but could this be reliably
done with just 3 years of data? It seems better to use a "drift" in units of month-1, and the
original values of sigma and phi. When I do this, I get values pretty close to what is shown in Fig
5, but a little larger. For example, here is the calculation the way I think it has been done in Fig
5, for a "drift" of 1*10-4:
(1.96*8.58e-5/(1e-4)*sqrt((1+0.58)/(1-0.58)))^(2/3) = 2.1994,
and the way I suggest it probably should be done:
(1.96*1.7e-4/(1e-4/12)*sqrt((1+0.89)/(1-0.89)))^(2/3)/12 = 2.5144

*We are trying to stay consistent with the literature on this subject.  One way to think of the confusion in units is that drift, as expressed in units per year, is being compared with variability, which can be described by variability of monthly, quarterly, or annual data.  However the data are collected (monthly, quarterly or annual, the magnitude of variability and autocorrelation will change appropriately.  But, the number of years to detect is strongly based on this ratio (variability to drift) and length of time. We agree with the reviewer that he /she presents  an acceptable alternative way to express things.We have added text to show the values we use in each section and believe the results are now easily reproducible.*

5. The abstract states "For relative drift to be identified within 0.1% yr-1 uncertainty, the overlap for these two satellites would need to be 2.6 years", while Fig 5 suggests rather that ~2.6 years is needed to identify a drift of 10%! This is a big difference. If the 10% value is correct, it has a pretty substantial practical implication for the study—it seems unlikely that any reasonable overlap period (of a small number of years) will be able to do much to constrain drifts of any but the most egregious magnitude.

*We think the results are now more easily reproducible by inclusion of the values used and why.*
*Part of the problem with matching the abstract information with the Figure 5, is that they are*
*both working off of different "relative" numbers.  The text is now more clear in each case.  We*
*are balancing the fact that as scientists, we'd like to understand the relative drift in our data, for*
*instance the SOLSTIC data; in this mindset we want the drift relative to the mean of SOLSTIC.*
*But overlap periods only allow us to get insight into the relative drift and the amount of*
*information is based on the differences between the two datasets.  As such, the results are based*
*on the characteristics of the differenced data and we present results in Figure 5 based on the*
*relative percentage of the differenced data.  We strengthened the text to make this more clear.*
*We do want this paper to be useful and thank the reviewer sincerely for his/her careful checking*
*of our results.  We conclude that the errors are all due to the variety of calculations performed*
*and think that the strengthening of the text will make the paper more useful.*
Some specific comments
P1, l27: Perhaps pedantic, but it's the satellite missions that overlap, not the satellites themselves.
*Change is made.*

P1, l29: this seems to be a result from another study, not this study, so probably shouldn't be in
the abstract.  *We'd like to keep this one in.  This was work done for this study—in fact this was*
*the exact point of the collaboration-- and is directly important to futher application of this work.*
P1, l32: actually 6 months (5.9), see major comments.  *We have double checked this and have*
*corrected the text.*
P2, l48: another pedantic point: the missions should overlap, not the launches  *Thank you.*
*Change is made.*
P5, Fig 2: these temperatures must be for a specific altitude range?  *We now make it more clear*
*that MSU Channel 4 and AMSU Channel 9 and refer to the lower stratosphere:  "Both channels*
*are designed to observe the lower stratosphere."*
P5, Fig 2: the relevance of Fig 2 to this study is questionable. It shows differences between 2
merged datasets with the offset and drift removed. But the offset and drift estimation is
specifically the theme of this paper!  *We've now added text to clarify:  "These plots show the*
*variability in overlap is highly dependent on latitude, as is often the case with Earth*
*observations."*
P6, l8: "our" as in the authors', or more generally?  *"Our" has now been removed.*
P6, l20: what "model" is being discussed here?  *Clarification has been made: "model" has been*
*replaced with "climate model."*
P6, l32: why can only "small" problems be identified?  *Thank you.  The sentence has been*
*changed to "… and even small problems can be identified."*
P7, l18: "respect" is a unique word choice, and I don't know exactly what the authors mean by it.
Thank you.  *We have changed "respect" to "appropriately incorporate" so the sentence now*
*reads:  In order to appropriately analyze satellite observations, it is necessary to understand*
*and appropriately incorporate the available information on the pre-flight calibration of*
*instruments and in-flight expected behavior.*
P8, l8: So many atmospheric measurements show variability at time scales longer than the
annual cycle due to modes of internal variability (e.g., ENSO, NAO) or responses to external
forcings (like solar variations!). So I'd be careful about implying that one year of data will
"cover the full range".  *We fully agree.  We have adjusted the sentence to clarify the point:*
*While Earth observations often require a minimum of a one year overlap to cover the full range*
*of expected observations, such arbitrary criteria ignore longer timescale phenomena including*
*ENSO and NAO, and are impractical for covering a full 11-year solar cycle in a planned overlap*
*period.*
P9, l6: These 3 points are contained in a paragraph describing Fig 3: it would be nice of point 1.
connected the pre-flight calibration estimates with what is shown in Fig 3.  *We have now tied the*
*text more directly to the figure. We have also made clear that the corrections that were*

*understood were already made in the construction of the datasets.    In each case, evaluation of*
*how best to characterize the drift takes place. For the Solstice-SIM data overlap, we noted that*
*the differences between the two sets of data, showed lower variability than the ratio of the data,*
*indicating an offset could be modeled as an additive adjustment."*
P9, l9: It's perhaps impossible in this case to know whether there is a true drift between the
instruments, or a "multiplicative bias". This might be worth pointing out at some point. *Thank*
*you, we had checked this out, but hadn't included it in the paper.  We've added the following*
*text:*
1. *There are jumps in the time series related to spacecraft and instrument anomalies.*
*Significant events are identified in instrument and spacecraft housekeeping telemetry and*
*changes in behavior before and after these events can be characterized and corrected in*
*the timese. Expamples of these phenomena are seen if Figurue 3 where SOLSTICE*
*experienced a failure of the mechanism that changes the entrance slit from the solar to*
*the stellar mode on 27 January 2006. The slit was moved back into position for*
*continuous solar observations but did not return to the exact same position so the optical*
*path through the instrument changed and therefore disrupted the degradation corrections*
*and the wavelength scale. Similarly, a spacecraft safe-hold event on 14 May 2007 caused*
*the instruments to become very cold and significantly changed the SIM wavelength scale*
*and perhaps the transmission properties of the instrument. The change in the SIM*
*wavelength grid is apparent in the uncorrected data, but in Figure 3 the data are*
*interpolated onto a standard mission-length wavelength scale and does not appear as a*
*jump in this figure. The 2007 safe-hold event had little effect on the performance of the*
*SOLSTICE. The jump associated with the 2006 SOLSTICE slit anomaly has also been*
*corrected and the change in character seen SOLSTICE data at this time represents the*
*best compromise over the full wavelength range of the instrument.*

P9, l19: Does SIM also show a jump here? *We now address this point in the text:  The change in*
*the SIM wavelength grid is apparent in the uncorrected data, but in Figure 3 the data are*
*inter"polated onto a standard mission-length wavelength scale and does not appear as a jump in*
*this figure."*

P11, Fig 4: Is this the same data as in Fig 3, just shown now in monthly means? The wavelength
of the measurements should be mentioned.  *We have now made clear that Figure 3 shows the*
*same data. Wavelength is now mentioned in abstract, Figure 3 caption and four other locations..*
P12, l7: Actually, depending on how sampling is dealt with, monthly means can exacerbate
sampling differences between instruments compared to shorter-term averages. See, e.g., Toohey
et al., 2013. *Thank you, I spoke with Dr. Toohey at some length on this general issue of monthly*
*means and we jointly constructed the sentence, " Monthly averages can remove higher frequency*
*noise and some sampling match-up problems, but they can also obscure important details and*
*can often introduce their own biases, especially when sampling is irregular in time or space*
*(Toohey et al., 2013). " I want to thank the reviewer for pointing us in the direction of both that*
*paper and the equally valuable, Toohey and von Clarmann, 2013 paper.*

P12, l26: actually 6 months.  *We get 5.2 months.  I think the difference between 5.2 and 5.9 is*

*due to the rounding errors of us reporting sigma and phi to only two significant digits.  Partly*
*for this reason, we have updated the table in the appendix to include more significant digits.*
P13: l29: This sentence, which includes Eq 3, is not grammatically correct.  *Fixed.  Thank you.*
P14, l4: grammatical issue with sentence 2 of Figure caption.  *Fixed.  Thank you.*